# ConFlux: Multivariate Time Series in Flux, One Unified Forecast in Confluence

Shiyu Wang [1] [*]   Juntong Ni [1] [*]   Ziyi Zhang [1]   Baichuan Mo [1]   Xinyue Zhong [1]   Chengxin Wang [1]
Yuchen Fang [1] [†]   Zhou Ye [1]   Yang Xiang [1]

## Abstract

Real-world multivariate time series are inherently in flux: different variables evolve asynchronously and interact in complex, time-varying ways, yet accurate forecasting requires these dispersed signals to converge into a single unified prediction. This structural mismatch between dynamic, heterogeneous inputs and a unified forecasting objective poses a fundamental challenge for building general-purpose multivariate forecasting models, especially in zero-shot and large-scale settings. To this end, inspired by the idea that "*all rivers run into the sea*", we propose **ConFlux**, a *general-purpose foundation model for multivariate time-series forecasting* by learning to adaptively integrate cross-channel information under a unified forecasting objective. Specifically, ConFlux first reorders variables to reduce cross-variable entanglement, then aggregates adjacent variables into compact patches that can be processed by a Vision Transformer-style architecture. This design shortens the effective context, reduces attention complexity, and provides a unified token representation for pre-training and downstream tasks. Experiments on 25 public datasets show that ConFlux achieves state-of-the-art performance in zero-shot, fine-tuning, and from-scratch settings, while offering faster inference and lower memory usage.

## 1. Introduction

Multivariate time series forecasting is central to understanding and predicting the behavior of dynamic systems across domains such as energy, finance, and transportation (Liang et al., 2024; Liu et al., 2025a; Xu et al., 2026). As shown in Figure 1, inspired by the idea that "*all rivers run into the sea*", real-world multivariate time series are inherently

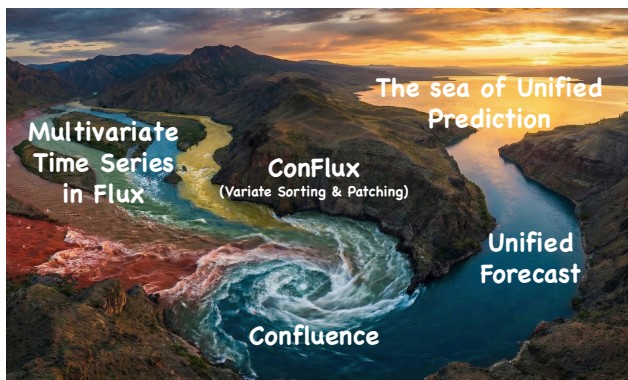

*Figure 1.* **From flux to confluence**: the core idea of ConFlux for multivariate time series forecasting. Multivariate time series evolve as heterogeneous, asynchronous streams, and ConFlux integrates their interactions into one unified forecast.

in flux: different variables evolve asynchronously, interact with each other, and form complex, time-varying structures. Rather than aligned channels, these variables resemble heterogeneous streams that twist, diverge, and intertwine over time. Yet effective forecasting ultimately requires these dispersed signals to converge into a single unified forecast. This tension between dynamic, heterogeneous inputs and a unified forecasting output lies at the heart of multivariate time series modeling and poses a fundamental challenge for building general-purpose forecasting models, particularly in zero-shot and large-scale settings.

This challenge becomes particularly pronounced in the context of foundation models. Recent deep learning models achieve strong performance when trained for specific datasets or tasks (Zhou et al., 2021; Wu et al., 2021), but require separate training for each domain, leading to high computational and storage cost. To address this limitation, recent work has proposed foundation models for univariate time series that generalize across tasks and datasets without task-specific retraining (Jin et al., 2024; Shi et al., 2025; Ansari et al., 2025). These models are effective in zero-shot settings largely because single-sequence inputs exhibit a relatively consistent structure across domains. However, this structural simplicity does not extend to multivariate signals, where interacting variables remain in flux (Qiu et al., 2025a; Fang et al., 2025a;b). Although specialized models for multivariate time series exist, extending them into

[*]Equal contribution [1]ByteDance. Correspondence to: Yuchen Fang <fyclmiss@gmail.com>.

*Proceedings of the 43rd International Conference on Machine Learning*, Seoul, South Korea. PMLR 306, 2026. Copyright 2026 by the author(s).

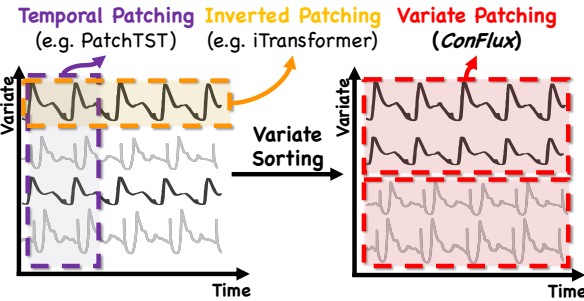

*Figure 2. **Variate patching**, a new modeling paradigm for multivariate time series used by ConFlux. Unlike temporal patching (Nie et al., 2023) and inverted patching (Liu et al., 2024b), it groups adjacent variates after **variate sorting** into patch tokens to reduce token count and support scalable cross-variate modeling.*

general-purpose foundation models remains challenging. In particular, scalable multivariate modeling faces two major obstacles. First, the quadratic complexity of self-attention across multiple variables imposes significant memory and computational costs during both training and inference (Liu et al., 2024b; Wang et al., 2024; Xue et al., 2023; Zhang & Yan, 2023). Second, specialized multivariate-oriented models often rely on fixed-length decoders (Ni et al., 2025; Zhao et al., 2025; Ekambaram et al., 2023), which limits their ability to perform flexible forecasting across varying prediction horizons. Addressing these challenges is essential for enabling general-purpose multivariate foundation models that combine accuracy, scalability, and adaptability.

To address these challenges of scalable multivariate forecasting, it is instructive to revisit how existing models reduce complexity in related domains. Drawing inspiration from Vision Transformers (Dosovitskiy, 2020; Han et al., 2022) and univariate models like PatchTST (Nie, 2022), patching mechanisms have become a prevalent strategy for lowering Transformer complexity by aggregating local information into compact tokens. In these contexts, the underlying assumption holds that *nearby elements—whether pixels in an image or adjacent time steps in a sequence—tend to share similar structure*, allowing a coarse representation to retain most of the signal's predictive content. However, this intuition does not directly extend to the variable dimension of multivariate time series: different variables may represent heterogeneous phenomena with arbitrary orderings and diverse interaction patterns. Simply patching variables without regard to their relationships can obscure important dependencies and degrade forecasting accuracy.

Motivated by this observation, we propose ConFlux, a purely multivariate transformer-based foundation model for time series forecasting. As shown in Figure 2, different from temporal patching (Nie, 2022) and inverted patching (Liu et al., 2024b), ConFlux introduces a *variate sorting* mechanism that sorts variables according to a comprehensive correlation matrix such that highly related variates are

placed adjacently. Once variables are ordered, ConFlux applies *variate patching* to aggregate adjacent, highly related variates into compact tokens. This reduces the number of input tokens to the Transformer encoder and thereby lowers attention complexity. To support flexible forecasting across arbitrary horizons—beyond the fixed prediction lengths of many prior architectures—we further employ a Fourier basis decoder, which reconstructs future values from historical representations corresponding to sine and cosine basis functions. This combination enables ConFlux to perform accurate, scalable multivariate forecasting while maintaining computational efficiency and adaptability across diverse tasks and domains.

In summary, this work makes the following contributions.

- To the best of our knowledge, we take an initial yet critical step toward exploring a purely variate transformer-based foundation model ConFlux for multivariate time series forecasting, which can capture cross-variable dependencies and perform efficient inference across variates.

- ConFlux introduces a novel architecture that combines a variate-patching Transformer encoder with a Fourier-basis decoder, supporting scalable modeling with an arbitrary number of input variates and flexible forecasting over arbitrary prediction horizons.

- We conduct extensive experiments on 11 public multivariate datasets and 14 large-scale dimensional datasets, demonstrating that ConFlux achieves state-of-the-art performance in zero-shot, fine-tuning, and from-scratch settings, while providing superior inference efficiency and reduced memory consumption.

## 2. Related Work

### 2.1. Multivariate Time Series Forecasting

Time series forecasting has long been a central task in deep learning. In practical settings, however, data are often multivariate, with many interacting signals whose joint evolution should be modeled to achieve accurate forecasting. This has motivated numerous architectures tailored to multivariate systems (Cui et al., 2021; Shao et al., 2022; Yi et al., 2023; Zhang & Yan, 2023; Wu et al., 2023b; Qiu et al., 2025d; 2026). Among them, iTransformer (Liu et al., 2024b) introduced the idea of inverting the Transformer's tokenization, embedding variable histories as attention tokens to capture interactions between channels. Complementary to this, TSMixer (Ekambaram et al., 2023) adopts an MLP-based mixer architecture that combines temporal and cross-series information through stacked mixing layers. To improve computational efficiency in multivariate forecasting, recent work such as SOFTS (Han et al., 2024) proposes

a series-core fusion mechanism that aggregates information into a global core and redistributes it, achieving linear complexity while maintaining competitive accuracy. Methods like LIFT (Zhao & Shen, 2025) further explore the use of leading indicator variables to guide prediction across variates, improving both efficiency and generalization across tasks. Beyond these architectural and complexity-oriented advances, clustering have also been used to refine multivariate modeling. DUET (Qiu et al., 2025c) introduces soft variate clustering to handle complex inter-variate relationships. Similarly, TimeFilter (Hu et al., 2025) applies patch-specific filtration to adaptively preserve the most relevant dependencies among variates, reducing noise and enhancing generalization. These diverse strands of research provide context for our work on general-purpose foundation models for multivariate time series.

## 2.2. Time Series Foundation Models

Time series foundation models (TSFMs) are large-scale pre-trained models designed to learn universal representations across diverse domains, enabling zero-shot or few-shot generalization to downstream forecasting and analysis tasks. Currently, time series foundation models can be broadly grouped into two main categories. The first category consists of LLM-based approaches, which adapt large language models to time series data (Jin et al., 2024; Liu et al., 2024a; 2025a). Because natural language lacks an explicit variate-modeling capability, these methods typically adopt a channel-independent strategy during fine-tuning and inference. The second category comprises time series pre-trained models that aim to build representations directly over time series data (Wang et al., 2025a;b; Shao et al., 2025; Liu et al., 2025b). Many of these also employ a channel-independent strategy to bypass the complexity of cross-variate interactions. Models such as Timer (Liu et al., 2024c), Chronos (Ansari et al., 2024), Time-MoE (Shi et al., 2025) fall into this group and have demonstrated strong predictive performance while remaining robust to varying variate counts. In contrast, other pre-trained methods explicitly incorporate variate correlation into the learning process. For instance, MOIRIA (Woo et al., 2024) and Timer-XL (Liu et al., 2025c) flattens all channels into a single sequence and differentiates them using positional embeddings, allowing self-attention to jointly capture temporal and cross-variate relationships. UniTS (Gao et al., 2024) and Chronos-2 (Ansari et al., 2025) take a related but distinct approach by applying self-attention over the variate dimension, directly modeling variate correlations during pre-training. Despite progress in capturing variate relationships, existing foundation models have not fully addressed the high computational cost of variate transformer. The quadratic complexity results in substantial memory overhead and slow inference, which severely constrains scalability to multivariate settings.

## 3. Problem Definition

### 3.1. General Multivariate Time Series Forecasting

We consider the problem of multivariate time series forecasting. Let $\mathbf{X} = \{\mathbf{x}_1, \mathbf{x}_2, \ldots, \mathbf{x}_T\} \in \mathbb{R}^{T \times C}$ denote a multivariate time series of length $T$ with $C$ variables, where $\mathbf{x}_t \in \mathbb{R}^C$ represents the observations of all variables at time step $t$. Given a historical context window of length $L$, the goal is to predict future values over a horizon of length $H$, by learning a forecasting function

$$
\begin{aligned}
f_\theta : \mathbf{X}_{1:L} &= \{\mathbf{x}_1, \ldots, \mathbf{x}_L\} \in \mathbb{R}^{L \times C} \to \\
\hat{\mathbf{X}}_{L+1:L+H} &= \{\hat{\mathbf{x}}_{L+1}, \ldots, \hat{\mathbf{x}}_{L+H}\} \in \mathbb{R}^{H \times C},
\end{aligned}
\tag{1}
$$

where $\theta$ denotes the model parameters.

Let $\mathcal{D} = \{\mathcal{X}^{(1)}, \ldots, \mathcal{X}^{(N)}\}$ denote a collection of multivariate time-series datasets, where each $\mathcal{X}^{(i)} = \{\mathbf{x}_1^{(i)}, \ldots, \mathbf{x}_{T_i}^{(i)}\} \in \mathbb{R}^{T_i \times C_i}$ may vary in sequence length $T_i$ and number of variables $C_i$ across domains. General forecasting aims to forecast future values for diverse downstream datasets and tasks without task-specific retraining:

$$
\begin{aligned}
f_\theta : \mathbf{X}_{1:L_i}^{(i)} &= \{\mathbf{x}_1^{(i)}, \ldots, \mathbf{x}_{L_i}^{(i)}\} \in \mathbb{R}^{L_i \times C_i} \to \\
\hat{\mathbf{X}}_{L_i+1:L_i+H_i}^{(i)} &= \{\hat{\mathbf{x}}_{L_i+1}^{(i)}, \ldots, \hat{\mathbf{x}}_{L_i+H_i}^{(i)}\} \in \mathbb{R}^{H_i \times C_i},
\end{aligned}
\tag{2}
$$

where $\theta$ denotes the model parameters.

## 4. Methodology

In this section, we describe the Conflux, a general-purpose foundation model for multivariate time series forecasting that is both scalable and efficient. As illustrated in Figure 3, Conflux is composed of four major components that work together to capture complex temporal and cross-variable dependencies while enabling flexible prediction horizons and reduced computational cost. *First*, a variate sorting module computes multiple measures of inter-variable correlation and derives an ordering of channels that places strongly related variables adjacent to each other. This ordering alleviates information conflict when aggregating tokens. *Second*, a variate embedding module projects each sorted variable's historical context into a shared latent space, forming the basic token representation for downstream modeling. *Third*, a variate patching Transformer encoder reduces the number of input tokens by aggregating adjacent variables into compact patches and applies self-attention over patches to efficiently capture joint dependencies across channels. *Finally*, a Fourier basis decoder reconstructs forecasts of arbitrary length by combining learned frequency coefficients with sine and cosine basis functions. In the following subsections, we provide a detailed description of each component and the overall training objective.

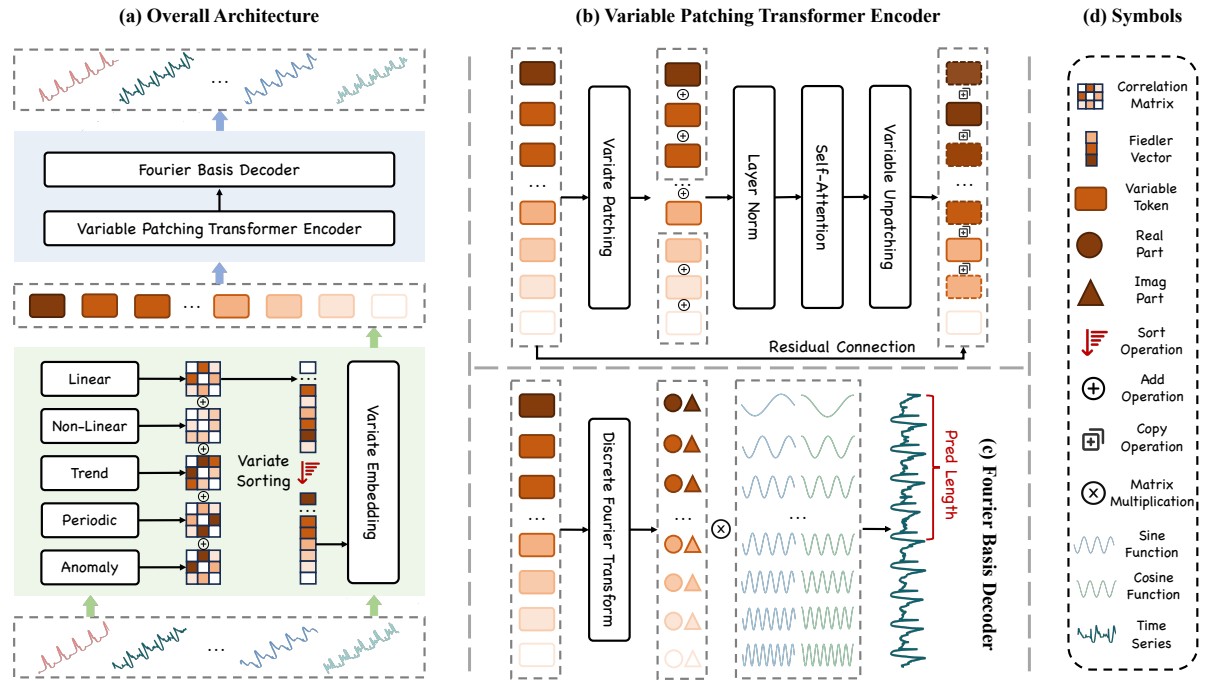

*Figure 3. (a) Overall architecture:* raw multivariate time series are first sorted and then embedded to form variate tokens; the variate tokens are processed by a Variate Patching Transformer Encoder, followed by a Fourier Basis Decoder to produce future values. *(b) Variate Patching Transformer Encoder:* variate tokens are patched, layer-normalized, and processed by self-attention; tokens are then unpatched with a residual connection. For clarity, the feed-forward network is omitted in the diagram but is used in the actual implementation. *(c) Fourier Basis Decoder:* sinusoidal bases (sine/cosine) are combined via learned real/imaginary components to generate predictions over the target horizon. *(d) Symbols:* legend of graphical notation.

## 4.1. Variate Sorting

To facilitate effective multivariate modeling under variate patching, we first sort variates such that highly correlated ones are placed adjacent. This design aims to mitigate information interference when aggregating multiple variates into patches. Given a multivariate time series $X \in \mathbb{R}^{L \times C}$, where $L$ is the context length and $C$ is the number of variables, we compute multiple pairwise association measures to capture diverse forms of inter-variable dependency. Specifically, we consider five complementary types of associations (Qiu et al., 2024), including linear correlation, non-linear dependence, trend similarity, periodic similarity, and anomaly co-occurrence (see Appendix A for details). Each measure yields an association matrix $A^{(k)} \in \mathbb{R}^{C \times C}$, $k = 1, \ldots, 5$, where $A_{i,j}^{(k)}$ reflects the strength of dependency between variables $i$ and $j$ under the $k$-th criterion. To obtain a unified representation of cross-variable relationships, we aggregate these association matrices by summation $A = \sum_{k=1}^{5} A^{(k)}$. The resulting matrix $A$ defines a weighted undirected graph over variables. We then compute the corresponding graph Laplacian $\mathcal{L}$ and extract its Fiedler vector (Fiedler, 1973), which corresponds to the eigenvector associated with the second smallest eigenvalue. Sorting variables according to the values of the Fiedler vector yields a one-dimensional ordering in which strongly related variables are placed closer

together. We provide a detailed theoretical guarantee of variate sorting in Appendix B. This ordering preserves local variable coherence and reduces information conflict when variables are grouped into patches.

## 4.2. Variate Embedding

After variate sorting, we embed each variate into a token representation suitable for Transformer processing. Inspired by the design of iTransformer (Liu et al., 2024b), we treat each variable as a token by projecting its entire historical window into a latent embedding space. Formally, for the sorted multivariate input $\tilde{X} \in \mathbb{R}^{L \times C}$, each variable sequence $\tilde{X}_{:,c}$ is embedded via a shared fully connected layer:

$$Z_{:,c} = W_e \tilde{X}_{:,c} + b_e, \ c = 1, \ldots, C, \tag{3}$$

where $W_e \in \mathbb{R}^{d \times L}$ and $b_e \in \mathbb{R}^d$. This produces a set of variate embeddings:

$$Z = \{Z_{:,1}, \ldots, Z_{:,c}\}^T \in \mathbb{R}^{C \times d}, \tag{4}$$

which serve as the input tokens to the subsequent model.

## 4.3. Model Architecture

### 4.3.1. VARIATE PATCHING TRANSFORMER ENCODER

**Variate Patching.** Directly applying self-attention over all $C$ variate tokens incurs quadratic complexity with respect to the number of variables, leading to high memory consumption and slow inference. To enable scalable multivariate foundation modeling, we introduce a variate patching Transformer encoder. We first partition the ordered variate tokens $Z$ into non-overlapping patches of size $P$, resulting in $N = \lceil \frac{C}{P} \rceil$ patches. Each patch is aggregated into a compact patch token via a average operation:

$$\bar{z}_i = \text{Mean}\left(z_{(i-1)P+1:(iP)}\right), \ i = 1, \ldots, N. \quad (5)$$

**Transformer Encoder.** These patch tokens are then processed by a standard Transformer encoder consisting of layer normalization, self-attention, and feed-forward blocks. Specifically, the Transformer block follows a pre-LayerNorm architecture: before self-attention and feed-forward blocks, LayerNorm normalizes per-token features, improving optimization stability for deep stacks. Multi-head self-attention then models cross-patch dependencies by computing $Q = \bar{Z}W_Q$, $K = \bar{Z}W_K$, $V = \bar{Z}W_V$ for each head, producing attention weights $\mathcal{A} \in \mathbb{R}^{C \times C}$ and outputs $\tilde{Z} \in \mathbb{R}^{C \times d}$:

$$\begin{aligned} \mathcal{A} &= \text{softmax}(QK^T/\sqrt{d}) \\ \tilde{Z} &= \mathcal{A}V \end{aligned} \quad (6)$$

By operating on $N \ll C$ tokens, variate patching reduces attention complexity from $O(C^2)$ to $O(N^2)$, substantially lowering memory requirements and accelerating both training and inference.

**Variate Unpatching.** After the self-attention, the patch representations are projected back to the variate level through an unpatching operation with residual connections, preserving fine-grained variable information.

### 4.3.2. FOURIER BASIS DECODER

To enable flexible forecasting over arbitrary horizons without modifying the core architecture, we adopt a Fourier basis driven decoder that reconstructs future time series using explicit sine and cosine basis functions. Our design is grounded in the observation that the real and imaginary parts of the discrete Fourier transform (DFT) correspond to the coefficients of cosine and sine basis functions at different frequency levels (Fang et al., 2024; Qiu et al., 2025b), respectively. This basis perspective reveals that the frequency components themselves carry interpretable time–frequency information when projected onto orthogonal sinusoidal bases. Given the encoded multivariate representation $\tilde{Z} \in \mathbb{R}^{C \times d}$, we first apply a DFT along the temporal

dimension to obtain complex frequency components:

$$\mathbf{F} = \text{DFT}(\tilde{Z}), \quad F_{c,k} = \text{Re}(F_{c,k}) + \text{i Im}(F_{c,k}), \quad (7)$$

where $F_{c,k} \in \mathbb{C}$ denotes the coefficient for variable $c$ at frequency bin $k$. From the basis functions perspective, the real part $\text{Re}(F_{c,k})$ and the imaginary part $\text{Im}(F_{c,k})$ serve as the amplitudes of cosine and sine basis functions at frequency $k$, respectively We then reconstruct the forecasted signal for each variable $c$ at an arbitrary future time index $t$ by combining these basis components:

$$\hat{z}_{t,c} = \sum_{k=0}^{K-1} \left(\text{Re}(F_{c,k}) \cdot \cos(\omega_k t) + \text{Im}(F_{c,k}) \cdot \sin(\omega_k t)\right), \quad (8)$$

where learnable $\omega_k$ denotes the discrete angular frequency of bin $k$ and $K$ is the number of considered frequency bins. In practice, the network learns to adjust the real and imaginary coefficients to best capture time–frequency patterns present in the training data.

By interpreting DFT coefficients as basis function amplitudes rather than raw spectral values, the decoder explicitly models the contribution of each sinusoidal component to future values. This representation preserves both time and frequency information and naturally supports forecasts of arbitrary length $H$ without requiring retraining or architectural constraints tied to fixed output sizes.

## 4.4. Training Objective

Let $Q = \{q_1, \ldots, q_{|Q|}\}$ denote a set of quantile levels for prediction. We adopt the standard quantile regression objective commonly used in time series forecasting (Woo et al., 2024; Ansari et al., 2025):

$$\sum_{q \in Q} \left[q \cdot \max\left(z - \hat{z}_q, \, 0\right) + (1 - q) \cdot \max\left(\hat{z}_q - z, \, 0\right)\right], \quad (9)$$

where $\hat{z}_q$ is the predicted value at quantile level $q$ and $z$ is the target value after normalization. This objective penalizes over- and under-predictions asymmetrically according to $q$, encouraging calibrated probabilistic forecasts.

In our architecture, Conflux's encoder and Fourier basis decoder produce output representations of shape $C \times H$ for each forecast horizon of length $H$. To support multi-quantile forecasting with the above loss, we first project the encoded representation $\tilde{Z} \in \mathbb{R}^{C \times d}$ into a quantile latent tensor $\mathbf{G} \in \mathbb{R}^{|Q| \times C \times d}$ with a learned linear layer:

$$\mathbf{G} = W_q \tilde{Z} + b_q, \quad (10)$$

where $W_q \in \mathbb{R}^{|Q| \times d}$ and $b_q \in \mathbb{R}^{|Q|}$ are quantile-specific parameters. The Fourier basis decoder then maps $\mathbf{G}$ to forecast $\hat{\mathbf{Z}} \in \mathbb{R}^{|Q| \times C \times H}$, with $\hat{z}_{q,c,h}$ representing the predicted value of variable $c$ at horizon step $h$ and quantile $q$.

*Table 1.* Zero-shot forecasting results of time series foundation models on long-term forecasting datasets. Results are reported on four forecasting horizons: $H \in \{96, 192, 336, 720\}$. The best and second-best results are highlighted in **bold** and underlined, respectively.

| Models | ConFlux Ours | | Chronos-2 2025 | | Toto 2025 | | Timer-XL 2025 | | Time-MoE 2025 | | Sundial 2025 | | TiRex 2025 | | FlowState 2025 | | Moirai 2024 | | Chronos 2024 | |
|---|---|---|---|---|---|---|---|---|---|---|---|---|---|---|---|---|---|---|---|---|
| Metrics | MSE | MAE | MSE | MAE | MSE | MAE | MSE | MAE | MSE | MAE | MSE | MAE | MSE | MAE | MSE | MAE | MSE | MAE | MSE | MAE |
| ETTh1 | **0.378** | **0.407** | 0.413 | 0.424 | 0.452 | 0.427 | 0.419 | 0.426 | 0.404 | 0.417 | 0.411 | 0.434 | 0.450 | 0.433 | 0.422 | 0.432 | 0.417 | 0.419 | 0.591 | 0.468 |
| ETTh2 | **0.309** | **0.374** | 0.344 | 0.389 | 0.370 | 0.383 | 0.347 | 0.388 | 0.366 | 0.404 | 0.333 | 0.387 | 0.348 | 0.374 | 0.354 | 0.380 | 0.362 | 0.382 | 0.405 | 0.410 |
| ETTm1 | **0.314** | **0.358** | 0.337 | 0.360 | 0.403 | 0.388 | 0.373 | 0.392 | 0.394 | 0.415 | 0.336 | 0.377 | 0.397 | 0.374 | 0.349 | 0.380 | 0.406 | 0.385 | 0.645 | 0.500 |
| ETTm2 | **0.242** | **0.306** | 0.245 | 0.307 | 0.276 | 0.313 | 0.273 | 0.336 | 0.317 | 0.365 | 0.258 | 0.320 | 0.290 | 0.321 | 0.256 | 0.308 | 0.311 | 0.337 | 0.310 | 0.350 |
| Weather | **0.229** | **0.263** | 0.269 | 0.282 | 0.248 | 0.273 | 0.256 | 0.294 | 0.265 | 0.297 | 0.234 | 0.270 | 0.269 | 0.276 | 0.258 | 0.295 | 0.281 | 0.287 | 0.279 | 0.306 |
| Solar | **0.162** | 0.209 | 0.170 | **0.206** | 0.492 | 0.271 | 0.771 | 0.604 | 0.411 | 0.428 | 0.221 | 0.245 | 0.222 | 0.214 | 0.746 | 0.727 | 0.714 | 0.704 | 0.393 | 0.319 |
| Electricity | **0.165** | 0.269 | 0.280 | **0.248** | 0.189 | 0.271 | 0.174 | 0.278 | - | - | 0.169 | 0.265 | 0.269 | 0.313 | 0.218 | 0.314 | 0.187 | 0.274 | 0.214 | 0.278 |

*Table 2.* Zero-shot forecasting results of time series foundation models on short-term forecasting datasets. Results are averaged across four forecasting horizons: $H \in \{12, 24, 48, 96\}$.

| Models | ConFlux Ours | | Chronos-2 2025 | | FlowState 2025 | | Chronos 2024 | |
|---|---|---|---|---|---|---|---|---|
| Metrics | MSE | MAE | MSE | MAE | MSE | MAE | MSE | MAE |
| PEMS03 | **0.085** | **0.190** | 0.132 | 0.219 | 0.368 | 0.395 | 0.136 | 0.229 |
| PEMS04 | **0.090** | **0.196** | 0.137 | 0.234 | 0.375 | 0.404 | 0.160 | 0.257 |
| PEMS07 | **0.071** | **0.173** | 0.117 | 0.214 | 0.320 | 0.374 | 0.172 | 0.233 |
| PEMS08 | **0.132** | **0.197** | 0.152 | 0.240 | 0.370 | 0.396 | 0.173 | 0.257 |

# 5. Experiment

## 5.1. Training Corpus.

We constructed a large-scale, cross-domain training corpus to facilitate the development of a multivariate time series foundation model, integrating various open-source datasets to ensure broad generalization. Specifically, we collect six large-scale real-world datasets: LargeST (Liu et al., 2023) from the transportation domain, Meter (Ni et al., 2025) and Wind (Wu et al., 2023b) from the energy domain, Air Quality (Ni et al., 2025) and ERA5 (Rasp et al., 2024) from the climate domain, and Atec (Ni et al., 2025) from the internet. These datasets vary in temporal dynamics, sampling frequencies, number of channels, and cross-variable dependencies, providing rich training signals for learning a shared representation across heterogeneous systems. In addition to real data, we augment the corpus with synthetic time series generated using the KernelSynth method introduced in Chronos (Ansari et al., 2024). The produced simulated sequences contain controllable temporal and multivariate dependency structures and enable the model to see a broad range of dependency patterns during training. This augmentation helps improve the generalization of the foundation model to unseen scenarios and dependency regimes.

## 5.2. Experimental Setting.

For evaluation, we select a broad set of benchmarks that cover long-term (Zhou et al., 2021; Wu et al., 2021; 2020; Godahewa et al., 2021), short-term (Song et al., 2020), and dimension-scaling (Ni et al., 2025) forecasting tasks (see Appendix D). As baselines, we compare our method with a diverse range of state-of-the-art models, grouped into two categories: *time series foundation models* and *models trained from scratch* (see Appendix E). Metric definitions are given in Appendix F, and implementation details are provided in Appendix G.

## 5.3. Performance Comparison

**Zero-Shot Forecasting.** To ensure a fair comparison, we evaluate all models in a zero-shot setting, where no in-domain fine-tuning is performed on downstream datasets. On *long-term forecasting datasets* as shown in Table 1, ConFlux outperforms competitive multivariate-capable baselines, including Chronos-2, Toto, Timer-XL, and Moirai, on the majority of datasets. Notably, ConFlux achieves an average reduction of **12.5%** in MSE compared to the strongest baseline, Chronos-2, with performance gains reaching as high as **41.1%** on the Electricity dataset, these results demonstrate ConFlux's exceptional zero-shot generalization and resilience to distribution shifts. In contrast, Toto, which introduces variable attention only at the last layer, is generally weaker, suggesting that late-stage cross-variable interaction is insufficient for zero-shot transfer in long-horizon settings. Models that flatten variables into a single sequence, such as Timer-XL and Moirai, also tend to underperform, which we attribute to the difficulty of preserving structured cross-variable dependencies under a flattened representation. On *short-term traffic forecasting* (Table 2), ConFlux consistently outperforms all baselines, achieving an average MSE reduction of **29.7%** over Chronos-2, with gains peaking at **39.3%** on PEMS07. This performance demonstrates ConFlux's superior ability to capture dense cross-variable

*Table 3.* Ablation study.

| Datasets | ETTh1 | | ETTh2 | | ETTm1 | | ETTm2 | | Weather | | Solar | | Electricity | |
|---|---|---|---|---|---|---|---|---|---|---|---|---|---|---|
| **Metrics** | MSE | MAE | MSE | MAE | MSE | MAE | MSE | MAE | MSE | MAE | MSE | MAE | MSE | MAE |
| w/o Sort | 0.397 | 0.414 | 0.329 | 0.386 | 0.325 | 0.364 | 0.247 | 0.309 | 0.240 | 0.273 | 0.165 | 0.212 | 0.167 | 0.271 |
| w/o PiT | 0.418 | 0.432 | 0.362 | 0.411 | 0.366 | 0.394 | 0.254 | 0.324 | 0.270 | 0.294 | 0.177 | 0.233 | 0.171 | 0.275 |
| w/o PiI | 0.390 | 0.410 | 0.312 | 0.374 | 0.334 | 0.369 | 0.257 | 0.314 | 0.236 | 0.268 | 0.163 | 0.209 | 0.165 | 0.269 |
| ConFlux | 0.378 | 0.407 | 0.309 | 0.374 | 0.314 | 0.358 | 0.242 | 0.306 | 0.229 | 0.263 | 0.162 | 0.209 | 0.165 | 0.269 |

*Table 4.* Forecasting results with in-domain fine-tuning of time series foundation models on long-term forecasting datasets.

| Models | ConFlux Ours | | Timer-XL 2025 | | Time-MoE 2025 | | Moirai 2024 | | Chronos 2024 | |
|---|---|---|---|---|---|---|---|---|---|---|
| Metrics | MSE | MAE | MSE | MAE | MSE | MAE | MSE | MAE | MSE | MAE |
| ETTh1 | **0.363** | **0.397** | 0.410 | 0.426 | 0.391 | 0.422 | 0.418 | 0.427 | 0.422 | 0.411 |
| ETTh2 | **0.305** | 0.371 | 0.354 | 0.396 | 0.353 | 0.395 | 0.339 | 0.394 | 0.343 | **0.369** |
| ETTm1 | **0.298** | **0.350** | 0.366 | 0.396 | 0.329 | 0.377 | 0.362 | 0.376 | 0.352 | 0.361 |
| ETTm2 | **0.205** | **0.285** | 0.281 | 0.343 | 0.297 | 0.348 | 0.261 | 0.323 | 0.263 | 0.300 |
| Weather | **0.206** | **0.246** | 0.247 | 0.291 | 0.236 | 0.284 | 0.274 | 0.304 | 0.268 | 0.278 |
| Solar | **0.148** | **0.199** | 0.601 | 0.524 | 0.423 | 0.344 | 0.450 | 0.458 | 0.187 | 0.288 |
| Electricity | 0.159 | 0.253 | 0.173 | 0.270 | - | - | 0.184 | 0.276 | **0.154** | **0.240** |

correlations and local dynamics. By effectively leveraging multivariate inductive biases, ConFlux ensures robust zero-shot generalization without requiring in-domain fine-tuning.

**Forecasting with In-domain Fine-tuning.** To assess the benefit of domain adaptation, we evaluate an in-domain fine-tuning setting on long-term forecasting benchmarks (Table 4). Following prior practice, we apply a consistent downstream protocol to all selected methods. Overall, Con-Flux remains the top performer after fine-tuning, achieving a **30.7%** average MSE reduction compared to the multivariate baseline Timer-XL. Notably, ConFlux consistently improves over its performance—notably yielding a **27.1%** gain on ETTm2—indicating its ability to effectively absorb domain-specific information without sacrificing stability. Compared with the multivariate flattening approach (Timer-XL), several univariate baselines benefit more from fine-tuning and become more competitive once exposed to in-domain data. This suggests that, given downstream data, optimizing for a coherent temporal structure can be more impactful than relying on cross-variable modeling obtained from flattening sequences. We attribute this to in-domain supervision emphasizing consistent temporal motifs and seasonality specific to the target domain. In this regime, refining time-aligned patterns becomes a primary driver of improvement, while naive multivariate mixing may introduce additional modeling burden not always rewarded by limited data. This aligns with our design choice of a Fourier basis decoder, providing an explicit mechanism to adjust periodic components. Consequently, ConFlux leverages in-domain signals to sharpen domain-specific temporal regularities, leading to

robust gains under fine-tuning.

Beyond the above benchmarks, we broaden the scope of our analysis by evaluating ConFlux under a from-scratch training setting (Appendix H) and in large-scale dimensional forecasting experiments (Appendix I).

### 5.4. Ablation Study

To better understand the contributions of the key design components in ConFlux, we conduct a comprehensive ablation study on six representative long-term forecasting benchmarks. Specifically, we evaluate the following variants:

- **w/o Sort**: without the variate sorting module.
- **w/o PiT**: without variate patching during training.
- **w/o PiI**: without variate patching during inference.

Table 3 summarizes the performance of ConFlux under these variants. Across all datasets and metrics, the full ConFlux model consistently outperforms its ablated counterparts. Removing variate sorting (w/o Sort) leads to noticeable degradation in both MSE and MAE, demonstrating that sorting variables according to inter-variable correlation facilitates more effective joint modeling. Omitting variate patching during training (w/o PiT) further degrades performance, indicating that patching not only improves efficiency but also serves as a useful inductive bias during representation learning. When variate patching is disabled only at inference (w/o PiI), performance also suffers relative to the full model, suggesting that the patching strategy contributes to robust and coherent cross-variable reasoning at test time. These results corroborate the importance of each component in ConFlux: variate sorting enhances the model's ability to capture cross-variable dependencies, and variate patching supports both scalable computation and accurate multivariate forecasting across diverse benchmarks.

### 5.5. Model Analysis

**Scalability Analysis** As a foundation model, it is important to understand how ConFlux's performance evolves with respect to both model capacity and training data scale. This analysis provides practical insights for effective deployment and fine-tuning under varying compute and data constraints. We evaluate four ConFlux variants with increasing parameter scales — Mini, Small, Base, and Large — trained on

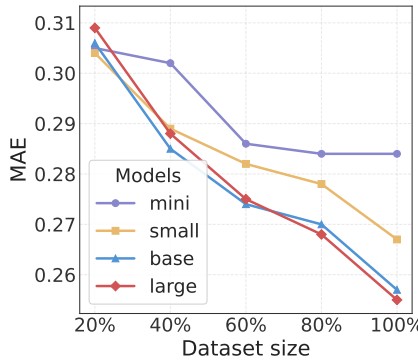

*Figure 4.* The scalability of ConFlux.

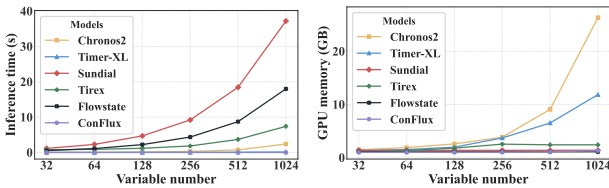

*Figure 5.* Inference efficiency comparison.

gradually larger fractions of the training corpus. Figure 4 shows the MAE curves for these variants as the proportion of training data increases. All model variants demonstrate consistent improvement as more data becomes available, confirming that ConFlux benefits from larger training corpora across model sizes. Notably, the curves exhibit different slopes: the larger models (Base and Large) show steeper declines in error with increasing data compared to the smaller variants (Mini and Small). This pattern indicates that higher-capacity ConFlux models are more effective at leveraging additional data to improve forecasting performance. These results highlight the scalability of ConFlux in two key aspects. First, increasing the amount of training data consistently enhances forecasting accuracy across all model sizes. Second, larger variants of ConFlux are able to extract greater benefit from additional data, demonstrating stronger data-scaling behavior. At the same time, compact variants remain competitive when data resources are limited, offering a flexible trade-off between efficiency and performance. Together, these findings underscore ConFlux's adaptability to diverse deployment scenarios with varying compute budgets and data availability.

**Inference Efficiency** To evaluate scalability, we measure inference efficiency of ConFlux under different numbers of variables and compare it with representative time series forecasting models. We report peak GPU memory usage and inference latency for variable counts from 32 to 1024, where lower values indicate better efficiency. As shown in Figure 5, ConFlux consistently achieves the best performance across all settings. Overall, ConFlux maintains low memory usage and low latency as the number of variables increases, while attention-based models show rapid growth

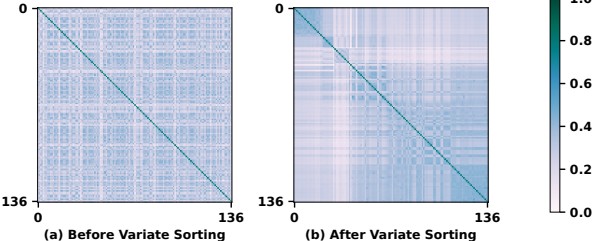

*Figure 6.* Effect of variate sorting on cross-variable relationship.

in both cost measures. Univariate models keep memory nearly constant but suffer from increasing latency due to repeated computation. In contrast, ConFlux scales smoothly in both memory and time, demonstrating strong practicality for large-scale multivariate forecasting. We provide more analysis in Appendix L.

**Visualizing the Effect of Variate Sorting** To better understand whether variate sorting truly makes the similar variates become closer, we visualize the aggregated association matrix $A$ of Solar before and after sorting in Figure 6. Each entry reflects the overall association strength between a pair of variables. Before sorting, strong dependencies are distributed across the matrix without clear locality. After sorting, variables with stronger mutual dependencies are placed adjacent, resulting in a more banded structure with higher similarities concentrated near the diagonal. This reorganization reveals locally coherent variable groups and reduces interference when adjacent variables are aggregated into patches. Together with the ablation results in Table 3, this visualization provides qualitative evidence that variate sorting effectively prepares inputs for variate patching. We also analyze the impact of patch size in Appendix J and the impact of look-back window length in Appendix K.

## 6. Conlusion

In this paper, we present ConFlux, a general-purpose foundation model for multivariate time series forecasting. To address the limitations of existing methods, ConFlux introduces a synergistic architecture combining variate sorting, variate patching, and a Fourier basis decoder. Specifically, our sorting mechanism reduces information conflict by organizing variables via refined correlations, while the patching strategy enables efficient modeling of cross-variable dependencies for an arbitrary number of inputs. Furthermore, the Fourier basis decoder ensures flexible inference across varied forecasting horizons. Extensive experiments on multiple benchmarks demonstrate that ConFlux consistently outperforms state-of-the-art foundation models in zero-shot scenarios and surpasses specialized models in fine-tuning settings, proving its robust generalization and efficiency.

## Impact Statement

This paper presents work whose goal is to advance the field of Machine Learning. There are many potential societal consequences of our work, none which we feel must be specifically highlighted here.

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

## A. Details of Five Complementary Association Measures

Given a multivariate time series $X \in \mathbb{R}^{L \times C}$, where $L$ denotes the context length and $C$ is the number of variables, we construct multiple pairwise association matrices to characterize diverse forms of inter-variate dependency. Specifically, we consider five complementary association measures, each yielding an association matrix $A^{(k)} \in \mathbb{R}^{C \times C}$, where $A_{ij}^{(k)}$ quantifies the dependency strength between variates $i$ and $j$ under the $k$-th criterion.

**Linear correlation** captures synchronous linear relationships between variates and is computed using Pearson correlation coefficients:

$$A_{i,j}^{(1)} = \frac{\sum_{t=1}^{L} (x_{i,t} - \bar{x}_i)(x_{j,t} - \bar{x}_j)}{\sqrt{\sum_{t=1}^{L} (x_{i,t} - \bar{x}_i)^2} \sqrt{\sum_{t=1}^{L} (x_{j,t} - \bar{x}_j)^2}}, \tag{11}$$

where

$$\bar{x}_i = \frac{1}{L} \sum_{t=1}^{L} x_{i,t}. \tag{12}$$

**Non-linear dependence** is quantified using mutual information, which captures complex and non-monotonic relationships beyond linear correlation:

$$A_{i,j}^{(2)} = I(X_i; X_j) = \sum_{x,y} p_{ij}(x,y) \log \frac{p_{ij}(x,y)}{p_i(x) \, p_j(y)}, \tag{13}$$

where $p_{ij}(x,y)$ and $p_i(x), p_j(y)$ denote the joint and marginal distributions, respectively, estimated via discretization.

**Trend similarity** reflects shared long-term dynamics between variates. We first extract trend components using a moving average filter,

$$\tilde{x}_{i,t} = \frac{1}{w} \sum_{k=0}^{w-1} x_{i,t-k}, \tag{14}$$

and then compute Pearson correlation between the resulting trend sequences:

$$A_{i,j}^{(3)} = \mathrm{Corr}(\tilde{X}_i, \tilde{X}_j). \tag{15}$$

**Periodic similarity** measures whether variates exhibit consistent seasonal or cyclic behaviors. For each variate, we extract the dominant period via Fourier analysis,

$$f_i^* = \arg \max_{f > 0} |\mathcal{F}(X_i)(f)|, \qquad p_i = \frac{1}{f_i^*}, \tag{16}$$

and define periodic similarity using a Gaussian kernel:

$$A_{i,j}^{(4)} = \exp\left(-\frac{(p_i - p_j)^2}{2\sigma_p^2}\right), \tag{17}$$

where $\sigma_p$ denotes the standard deviation of dominant periods across variates.

**Anomaly co-occurrence** captures shared extreme behaviors. We identify anomalous time points using z-score normalization,

$$z_{i,t} = \frac{x_{i,t} - \mu_i}{\sigma_i}, \qquad a_{i,t} = \mathbb{I}(|z_{i,t}| > \tau), \tag{18}$$

and compute the anomaly overlap ratio:

$$A_{i,j}^{(5)} = \frac{\sum_{t=1}^{L} a_{i,t} a_{j,t}}{\sum_{t=1}^{L} a_{i,t} + \sum_{t=1}^{L} a_{j,t}}. \tag{19}$$

Each association matrix is normalized to $[0, 1]$, and the final association matrix is obtained by weighted aggregation:

$$A = \frac{\sum_{k=1}^{5} w_k \tilde{A}^{(k)}}{\sum_{k=1}^{5} w_k}. \tag{20}$$

## B. Theoretical Analysis of Variate Sorting

**Objective.** Let $C \in \mathbb{N}$ be the number of variates. Let the aggregated association matrix $A \in \mathbb{R}^{C \times C}$ satisfy

$$A_{ij} = A_{ji} \geq 0.$$

Define $d_i := \sum_{j=1}^{C} A_{ij}$, $D := \text{diag}(d_1, \ldots, d_C)$, and the (unnormalized) graph Laplacian

$$L := D - A.$$

A linear ordering is represented by an integer position vector $y \in \{1, \ldots, C\}^C$ with the permutation constraint

$$\mathcal{P} := \left\{ y \in \{1, \ldots, C\}^C : \{y_i\}_{i=1}^{C} = \{1, \ldots, C\} \right\}.$$

We seek *an ordering that places strongly associated variates close to each other* by minimizing the weighted squared positional differences:

$$\min_{y \in \mathcal{P}} \sum_{1 \leq i < j \leq C} A_{ij}(y_i - y_j)^2. \tag{21}$$

**1. Quadratic-form identity.** For any $y \in \mathbb{R}^C$,

$$\sum_{1 \leq i < j \leq C} A_{ij}(y_i - y_j)^2 = \frac{1}{2} \sum_{i,j=1}^{C} A_{ij}(y_i - y_j)^2 \tag{22}$$

$$= \frac{1}{2} \sum_{i,j=1}^{C} A_{ij}\left(y_i^2 - 2y_i y_j + y_j^2\right)$$

$$= \frac{1}{2} \sum_{i,j=1}^{C} A_{ij} y_i^2 - \sum_{i,j=1}^{C} A_{ij} y_i y_j + \frac{1}{2} \sum_{i,j=1}^{C} A_{ij} y_j^2$$

$$= \sum_{i=1}^{C} \left(\sum_{j=1}^{C} A_{ij}\right) y_i^2 - \sum_{i,j=1}^{C} A_{ij} y_i y_j \tag{23}$$

$$= \sum_{i=1}^{C} d_i y_i^2 - \sum_{i,j=1}^{C} A_{ij} y_i y_j$$

$$= y^\top D y - y^\top A y$$

$$= y^\top (D - A) y$$

$$= y^\top L y. \tag{24}$$

In (22), we used $A_{ij} = A_{ji}$ and $(y_i - y_j)^2 = (y_j - y_i)^2$, hence $\frac{1}{2} \sum_{i,j} A_{ij}(y_i - y_j)^2 = \sum_{i<j} A_{ij}(y_i - y_j)^2$ (the $i = j$ terms are zero). In (23), we used

$$\frac{1}{2} \sum_{i,j} A_{ij} y_i^2 + \frac{1}{2} \sum_{i,j} A_{ij} y_j^2 = \sum_{i} \left(\sum_{j} A_{ij}\right) y_i^2.$$

Therefore, (21) is equivalent to

$$\min_{y \in \mathcal{P}} \sum_{1 \leq i < j \leq C} A_{ij}(y_i - y_j)^2 \quad \Longleftrightarrow \quad \min_{y \in \mathcal{P}} y^\top L y. \tag{25}$$

**2. Centering.** Let

$$\mu := \frac{1}{C} \sum_{t=1}^{C} t = \frac{C+1}{2}, \qquad \rho^2 := \sum_{t=1}^{C} (t - \mu)^2 = \frac{C(C^2 - 1)}{12},$$

and define $z := y - \mu\mathbf{1}$. Then for all $y \in \mathcal{P}$,

$$z^\top \mathbf{1} = 0, \qquad \|z\|_2^2 = \rho^2.$$

Moreover, since $L\mathbf{1} = 0$,

$$y^\top L y = (z + \mu\mathbf{1})^\top L(z + \mu\mathbf{1}) = z^\top L z.$$

Define the centered discrete feasible set

$$\mathcal{Z} := \{y - \mu\mathbf{1} : y \in \mathcal{P}\}.$$

Thus (25) is equivalent to

$$\min_{z \in \mathcal{Z}} z^\top L z. \tag{26}$$

**3. Spectral relaxation.** Let

$$\mathcal{S} := \{z \in \mathbb{R}^C : z^\top \mathbf{1} = 0, \|z\|_2^2 = \rho^2\}.$$

Since $\mathcal{Z} \subseteq \mathcal{S}$,

$$\min_{z \in \mathcal{S}} z^\top L z \ \leq\ \min_{z \in \mathcal{Z}} z^\top L z.$$

Consider the relaxed problem

$$\min_{z \in \mathbb{R}^C} z^\top L z \quad \text{s.t.} \quad z^\top \mathbf{1} = 0, \ \|z\|_2^2 = \rho^2. \tag{27}$$

**Theorem B.1** (Spectral minimizer; Fiedler vector (Fiedler, 1973)). *Let $0 = \lambda_1 \leq \lambda_2 \leq \cdots \leq \lambda_C$ be the eigenvalues of $L$ with corresponding orthonormal eigenvectors $\{u_k\}_{k=1}^C$, where $u_1 = \mathbf{1}/\sqrt{C}$. Then any minimizer of (27) satisfies*

$$z^\star \in \{\pm\rho\, u_2\} \quad \text{(if $\lambda_2$ has multiplicity 1),}$$

*and the optimal value is $\lambda_2\rho^2$.*

*Proof.* For any feasible $z$ with $z^\top \mathbf{1} = 0$, expand $z = \sum_{k=2}^C \alpha_k u_k$ with $\sum_{k=2}^C \alpha_k^2 = \|z\|_2^2 = \rho^2$. Then

$$z^\top L z = \sum_{k=2}^C \alpha_k^2 \lambda_k \geq \lambda_2 \sum_{k=2}^C \alpha_k^2 = \lambda_2\rho^2,$$

with equality iff $z \in \text{span}(u_2)$. Enforcing $\|z\|_2 = \rho$ yields $z^\star = \pm\rho u_2$. $\square$

**4. Optimal projection onto permutations.** Define the Euclidean projection (rounding) of $z^\star$ onto $\mathcal{Z}$:

$$\hat{z} \in \arg\min_{z \in \mathcal{Z}} \|z - z^\star\|_2^2. \tag{28}$$

Since $\|z\|_2^2 = \|z^\star\|_2^2 = \rho^2$ for all $z \in \mathcal{Z}$,

$$\|z - z^\star\|_2^2 = 2\rho^2 - 2z^\top z^\star,$$

hence (28) is equivalent to

$$\hat{z} \in \arg\max_{z \in \mathcal{Z}} z^\top z^\star. \tag{29}$$

Write $z = y - \mu\mathbf{1}$ with $y \in \mathcal{P}$. Since $(z^\star)^\top \mathbf{1} = 0$,

$$z^\top z^\star = (y - \mu\mathbf{1})^\top z^\star = y^\top z^\star,$$

thus (29) becomes

$$\hat{y} \in \arg\max_{y \in \mathcal{P}} \sum_{i=1}^C y_i\, z_i^\star. \tag{30}$$

Let $i_1, \ldots, i_C$ be indices such that

$$z_{i_1}^\star \leq z_{i_2}^\star \leq \cdots \leq z_{i_C}^\star.$$

By the rearrangement inequality, a maximizer of (30) is obtained by assigning

$$\hat{y}_{i_t} = t, \qquad t = 1, \ldots, C, \tag{31}$$

up to permutations within tied entries (ties) and a global reversal (sign ambiguity of $u_2$). Equivalently, sorting the Fiedler vector $u_2$ induces an optimal Euclidean projection of the relaxed optimum onto the discrete feasible set $\mathcal{P}$.

**Conclusion.** The resulting ordering is the optimal rounding of the spectral relaxation of (21), and hence promotes small positional distances for pairs with large association weights $A_{ij}$.

## C. DFT as Cosine/Sine Basis Coefficients

In this appendix, we provide a formal justification for interpreting the real and imaginary parts of the discrete Fourier transform (DFT) as coefficients of cosine and sine basis functions. This interpretation is fundamental to our Fourier basis decoder design.

The DFT of a discrete time sequence $x[n]$ of length $N$ is defined as:

$$X[k] = \sum_{n=0}^{N-1} x[n]\, e^{-j \frac{2\pi}{N} kn}, \quad k = 0, 1, \ldots, N-1,$$

where $j = \sqrt{-1}$ denotes the imaginary unit. This expression can be viewed as the projection of the time-domain signal onto a set of complex exponentials $e^{-j \frac{2\pi}{N} kn}$, which form an orthogonal basis in the vector space of length-$N$ sequences.

Using Euler's identity,

$$e^{-j\theta} = \cos(\theta) - j\sin(\theta),$$

we expand the complex exponential in the DFT definition as:

$$X[k] = \sum_{n=0}^{N-1} x[n] \left( \cos\left( \frac{2\pi}{N} kn \right) - j\sin\left( \frac{2\pi}{N} kn \right) \right).$$

By separating the real and imaginary parts, we obtain:

$$\mathrm{Re}(X[k]) = \sum_{n=0}^{N-1} x[n] \cos\left( \frac{2\pi}{N} kn \right), \tag{32}$$

$$\mathrm{Im}(X[k]) = - \sum_{n=0}^{N-1} x[n] \sin\left( \frac{2\pi}{N} kn \right). \tag{33}$$

These identities show that the real part $\mathrm{Re}(X[k])$ is the inner product of $x[n]$ with the cosine basis function at frequency $\frac{2\pi}{N} k$, and the imaginary part $\mathrm{Im}(X[k])$ is the (negated) inner product with the corresponding sine basis function. In this sense, each DFT coefficient $X[k]$ encodes how strongly the signal correlates with the respective sinusoidal components.

To further illustrate how these coefficients are used to reconstruct the original signal, consider the inverse DFT (IDFT):

$$x[n] = \frac{1}{N} \sum_{k=0}^{N-1} X[k]\, e^{j \frac{2\pi}{N} kn}.$$

Substituting Euler's identity again yields:

$$x[n] = \frac{1}{N} \sum_{k=0}^{N-1} \left( \mathrm{Re}(X[k]) \cos\left( \frac{2\pi}{N} kn \right) - \mathrm{Im}(X[k]) \sin\left( \frac{2\pi}{N} kn \right) \right).$$

This expression explicitly reconstructs the time-domain signal as a linear combination of cosine and sine basis functions, with $\mathrm{Re}(X[k])$ and $\mathrm{Im}(X[k])$ serving as the respective coefficients.

Thus, the real and imaginary parts of the DFT coefficients correspond precisely to the amplitudes of cosine and sine basis functions at discrete frequency levels. This basis function perspective justifies our choice of using the real and imaginary components as basis weights in the Fourier basis decoder, enabling an explicit sinusoidal decomposition that supports flexible, arbitrary-length forecasting.

# D. Evaluation Datasets.

For evaluation, we select a comprehensive suite of inference datasets covering long-term, short-term, and high-dimensional forecasting tasks. For long-term forecasting, we use the widely adopted ETTh1, ETTh2, ETTm1, ETTm2 benchmarks (Zhou et al., 2021), as well as Weather (Wu et al., 2021), Solar (Wu et al., 2020), and ECL (Godahewa et al., 2021) datasets, which are standard in the literature for assessing multi-step predictive performance across different temporal granularities and domains. For short-term forecasting, we utilize four traffic speed datasets (Song et al., 2020): PEMS03, PEMS04, PEMS07, and PEMS08, which require the model to capture cross-variable dependencies. To evaluate performance in high-dimensional settings, we also include the datasets used in U-Cast (Ni et al., 2025), focusing on scenarios with a large number of variables. High-dimensional datasets are used primarily to assess training from scratch performance, as the training corpus already contains high-dimensional examples, ensuring a fair comparison.

# E. Baseline Details

We benchmark our approach against a diverse set of state-of-the-art models, categorized into two groups: **Time series foundation models** that are evaluated in fine-tuning and zero-shot settings. These include Chronos-2 (Ansari et al., 2025), Toto (Cohen et al., 2025), Timer-XL (Liu et al., 2025c), Time-MoE (Shi et al., 2025), Sundial (Liu et al., 2025d), TiRex (Auer et al., 2025), FlowState (Graf et al., 2025), Moirai (Woo et al., 2024), and Chronos (Ansari et al., 2024). These baselines are representative of recent efforts in building generalizable pre-trained models for time series tasks, enabling assessment of transferability and efficiency of learned representations. **Models trained from scratch**, which are designed for specific datasets or forecasting tasks. These include U-Cast (Ni et al., 2025), DUET (Qiu et al., 2025c), PAttn (Tan et al., 2024), CycleNet (Lin et al., 2024), TimeXer (Wang et al., 2024), SOFTS (Han et al., 2024), iTransformer (Liu et al., 2024b), PatchTST (Nie et al., 2023), Crossformer (Zhang & Yan, 2023), TimesNet (Wu et al., 2023a), DLinear (Zeng et al., 2023), and SCINet (Liu et al., 2022). These models represent a broad range of architectural choices, from attention-based and MLP-based designs to hybrid variants.

## E.1. Time Series Foundation Models

- Chronos-2: A foundation model that stacks temporal and variate attention blocks to jointly reason over time and cross-variable dependencies.

- Toto: A foundation model with a final variate attention layer focused on capturing inter-variable correlations.

- Timer-XL: A Transformer variant that flattens time and variate dimensions into a single sequence, enabling high-capacity joint modeling via a unified attention mechanism.

- Time-MoE: A single-variable autoregressive foundation model built on a mixture-of-experts architecture tailored for time series.

- Sundial: A patch-based single-variable foundation model that uses flow matching for training.

- TiRex: A classic single-variable LSTM-based time series foundation model.

- FlowState: A single-variable foundation model based on State Space Models for time series forecasting.

- Moirai: A variant that performs time and variate flattening similar to Timer-XL, integrating cross-series correlation via flattened attention.

- Chronos: A single-variable foundation model using discretized tokenization, representing time series as discrete tokens for foundation-scale learning.

## E.2. Time Series Specialized Models

- DUET: A multivariate specialized model employing dual clustering across time and variables to capture both temporal patterns and inter-channel correlations.

- CycleNet: A model tailored for cyclic behavior, emphasizing periodic/seasonal pattern modeling in univariate sequences.

- TimeXer: A specialized model that explicitly incorporates covariate modeling to improve forecast accuracy.

- SOFTS: A multivariate model that fuses all variables into a single anchor representation, enabling efficient modeling with reduced complexity.

- iTransformer: A Transformer-based baseline that performs full quadratic variate attention over channels.

- PatchTST: A patch Transformer model designed for single-variable time series, splitting long sequences into patches before attention for scalable modeling.

- Crossformer: A multivariate model built on a auto-correlation mechanism, designed to enhance cross-series and temporal interactions.

- TimesNet: A frequency-domain and convolutionally enhanced model that captures joint temporal patterns through frequency features and convolutional modules.

- DLinear: A simple yet effective univariate model based on direct historical-to-future linear mapping.

- SCINet: A model using 1D temporal convolution with hierarchical structure designed to capture multi-scale temporal dependencies within individual series.

These baselines span diverse modeling paradigms — from foundation models with varying attention and tokenization strategies to specialized architectures emphasizing temporal, frequency, and covariate structures. This comprehensive set enables rigorous evaluation of ConFlux across both general-purpose and task-specific forecasting scenarios.

*Table 5.* Training configurations of ConFlux.

| Model | Patch Size P | Context Length T | Prediction Length F | Layers L | MHA Heads d | MHA Dims $d_{ff}$ | FFN Dims H | Total Parameters # Count |
|---|---|---|---|---|---|---|---|---|
| ConFlux$_{Mini}$ | 16 | 2048 | 720 | 3 | 4 | 256 | 512 | 3M |
| ConFlux$_{Small}$ | 16 | 2048 | 720 | 6 | 6 | 336 | 768 | 9M |
| ConFlux$_{Base}$ | 16 | 2048 | 720 | 9 | 8 | 512 | 2048 | 29M |
| ConFlux$_{Large}$ | 16 | 2048 | 720 | 12 | 12 | 768 | 3072 | 87M |

## F. Metrics

We adopt multiple evaluation metrics to assess both predictive performance and computational efficiency. For forecasting accuracy, we report Mean Absolute Error (MAE) and Mean Squared Error (MSE), computed over target dimensions on forecast steps. These metrics are standard in time series forecasting and allow direct comparison with prior work. To measure model efficiency and scalability, we also evaluate inference speed (in seconds) and GPU memory usage (in megabytes) during prediction. Inference speed quantifies the time required to generate forecasts for a given test batch, while memory usage reflects the peak GPU memory consumption. These efficiency metrics are critical for practical deployment of foundation models in resource-constrained settings and for understanding the trade-offs between modeling complexity and real-world performance.

## G. Implementation Details

For all experiments, we adopt a consistent architecture and training configuration to ensure fair evaluation across datasets and baselines. The core of our model is a Transformer-based encoder composed of 9 layers, where each layer uses a model dimension of $d_{model} = 512$, a feed-forward network dimension of $d_{ff} = 2048$, and 8 attention heads. Under this configuration, the total number of trainable parameters in Conflux is approximately 29 million. We train the model on a machine equipped with eight NVIDIA B200 GPUs using a fixed learning rate of 0.001. Training proceeds for three epochs, with a per-GPU batch size of 32. To enable efficient and consistent processing of multivariate inputs with varying numbers of variables across datasets, we first preprocess the datasets by applying a variable patching step that unifies all datasets to have exactly 1000 variables. During training, the historical context length is set to 2048, allowing the model to observe long temporal dependencies. The patch size used in the variate patching Transformer encoder is 16.

We design and train four model configurations to reflect the scalability characteristics of ConFlux, from compact to large-capacity variants. As shown in Table 5, these configurations — Mini, Small, Base, and Large — differ systematically in their network dimensions, enabling controlled scaling of model capacity. These configurations cover a broad range of capacity levels, from lightweight architectures suitable for resource-limited scenarios to high-capacity models for data-rich settings. The observed performance trends reported in the scalability analysis (Section 6.4) reflect the impact of increasing representational capacity on forecasting accuracy and data utilization. For comparisons with baselines throughout the main experiments, we adopt the Base model as the default configuration. This choice strikes a favorable trade-off between performance and efficiency, and its parameter scale is consistent with most contemporary time series foundation models used as baselines. Using a common capacity level avoids unfair advantages arising purely from parameter count differences. Moreover, in all training settings, we follow the quantile regression setup of Chronos-2 by expanding the quantile set from 9 to 21 quantiles, ensuring robust distribution-aware forecasting. All other training hyperparameters — including optimizer settings, batch size, and learning rate schedules — are kept consistent across model sizes to isolate the effect of architectural scaling.

*Table 6.* Forecasting results with from-scratch training of long-term multivariate time series forecasting.

| Models | ConFlux Ours | | DUET 2025 | | CycleNet 2024 | | TimeXer 2024 | | SOFTS 2024 | | iTransformer 2024 | | PatchTST 2024 | | Crossformer 2023 | | TimesNet 2023 | | DLinear 2023 | | SCINet 2022 | |
|---|---|---|---|---|---|---|---|---|---|---|---|---|---|---|---|---|---|---|---|---|---|---|
| Metrics | MSE | MAE | MSE | MAE | MSE | MAE | MSE | MAE | MSE | MAE | MSE | MAE | MSE | MAE | MSE | MAE | MSE | MAE | MSE | MAE | MSE | MAE |
| ETTh1 | **0.430** | **0.428** | 0.443 | 0.436 | 0.457 | 0.441 | 0.437 | 0.437 | 0.449 | 0.442 | 0.454 | 0.448 | 0.469 | 0.455 | 0.529 | 0.522 | 0.458 | 0.450 | 0.456 | 0.452 | 0.747 | 0.647 |
| ETTh2 | **0.363** | **0.391** | 0.372 | 0.397 | 0.388 | 0.409 | 0.367 | 0.396 | 0.385 | 0.408 | 0.383 | 0.407 | 0.387 | 0.407 | 0.942 | 0.684 | 0.414 | 0.427 | 0.559 | 0.515 | 0.954 | 0.723 |
| ETTm1 | **0.379** | **0.385** | 0.390 | 0.393 | 0.386 | 0.395 | 0.382 | 0.397 | 0.393 | 0.403 | 0.407 | 0.410 | 0.387 | 0.400 | 0.513 | 0.495 | 0.400 | 0.406 | 0.403 | 0.407 | 0.486 | 0.481 |
| ETTm2 | **0.272** | 0.317 | 0.280 | 0.324 | 0.272 | **0.315** | 0.289 | 0.330 | 0.287 | 0.330 | 0.288 | 0.332 | 0.281 | 0.326 | 0.757 | 0.611 | 0.291 | 0.333 | 0.350 | 0.401 | 0.571 | 0.537 |
| Weather | **0.239** | **0.262** | 0.251 | 0.273 | 0.243 | 0.271 | 0.241 | 0.271 | 0.255 | 0.278 | 0.258 | 0.278 | 0.259 | 0.273 | 0.259 | 0.315 | 0.259 | 0.287 | 0.265 | 0.317 | 0.292 | 0.363 |
| Solar | **0.207** | 0.242 | 0.237 | **0.233** | 0.210 | 0.261 | 0.237 | 0.302 | 0.229 | 0.256 | 0.233 | 0.262 | 0.236 | 0.266 | 0.641 | 0.639 | 0.301 | 0.319 | 0.330 | 0.401 | 0.282 | 0.375 |
| Electricity | **0.159** | **0.250** | 0.172 | 0.258 | 0.168 | 0.259 | 0.171 | 0.270 | 0.174 | 0.264 | 0.178 | 0.270 | 0.205 | 0.290 | 0.244 | 0.334 | 0.193 | 0.295 | 0.212 | 0.300 | 0.571 | 0.537 |

*Table 7.* Forecasting results with from-scratch training of short-term multivariate time series forecasting.

| Models | ConFlux Ours | | DUET 2025 | | CycleNet 2024 | | TimeXer 2024 | | SOFTS 2024 | | iTransformer 2023 | | PatchTST 2023 | | Crossformer 2023 | | TimesNet 2022 | | DLinear 2022 | | SCINet 2022 | |
|---|---|---|---|---|---|---|---|---|---|---|---|---|---|---|---|---|---|---|---|---|---|---|
| Metrics | MSE | MAE | MSE | MAE | MSE | MAE | MSE | MAE | MSE | MAE | MSE | MAE | MSE | MAE | MSE | MAE | MSE | MAE | MSE | MAE | MSE | MAE |
| PEMS03 | **0.100** | **0.205** | 0.103 | 0.212 | 0.118 | 0.226 | 0.112 | 0.214 | 0.104 | 0.210 | 0.113 | 0.222 | 0.180 | 0.291 | 0.169 | 0.282 | 0.147 | 0.248 | 0.278 | 0.375 | 0.114 | 0.224 |
| PEMS04 | **0.079** | **0.181** | 0.106 | 0.217 | 0.119 | 0.232 | 0.105 | 0.209 | 0.102 | 0.208 | 0.111 | 0.221 | 0.195 | 0.307 | 0.209 | 0.314 | 0.129 | 0.241 | 0.295 | 0.388 | 0.093 | 0.202 |
| PEMS07 | **0.070** | **0.167** | 0.090 | 0.196 | 0.113 | 0.214 | 0.085 | 0.182 | 0.087 | 0.184 | 0.101 | 0.204 | 0.211 | 0.303 | 0.235 | 0.315 | 0.124 | 0.225 | 0.329 | 0.396 | 0.119 | 0.217 |
| PEMS08 | **0.095** | **0.197** | 0.114 | 0.217 | 0.150 | 0.246 | 0.175 | 0.250 | 0.138 | 0.219 | 0.150 | 0.226 | 0.280 | 0.321 | 0.268 | 0.307 | 0.193 | 0.271 | 0.379 | 0.416 | 0.159 | 0.244 |

## H. Forecasting with From-scratch Training.

We further evaluate a from-scratch training setting, where all models are trained on the target datasets under the same forecasting protocol. As shown in Table 6 and Table 7, the results indicate that ConFlux is not only effective in transfer settings, but also strong when trained directly on the target domain. Comparing against our model, Crossformer and iTransformer are generally less stable across datasets. This indicates that fully dense variable dependency modeling can be data-hungry and sensitive to domain heterogeneity in the from-scratch regime, where the training set may not be sufficient to reliably estimate all pairwise interactions. In contrast, SOFTS uses an anchor mechanism to aggregate variables, tends to be more competitive than fully dense baselines, supporting the view that structured coupling can mitigate overfitting to spurious correlations and DUET performs soft clustering over variables, is also consistently strong, suggesting that learning group-level dependencies is an effective compromise between expressiveness and statistical efficiency.

## I. Large-Scale Dimensional Experiment.

As shown in Table 8, ConFlux achieves strong and stable performance across high-dimensional datasets, often obtaining the best or second-best results under from-scratch training. While several baselines suffer accuracy drops or run out of

*Table 8.* Results of high-dimensional multivariate time series forecasting, where '—' indicates that the model ran out of memory.

| Methods | DLinear (2023) | | PatchTST (2023) | | PAttn (2024) | | TimesNet (2023) | | TSMixer (2023) | | iTransformer (2024) | | DUET (2025) | | U-CAST (2025) | | ConFlux (ours) | |
|---|---|---|---|---|---|---|---|---|---|---|---|---|---|---|---|---|---|---|
| Metrics | MSE | MAE | MSE | MAE | MSE | MAE | MSE | MAE | MSE | MAE | MSE | MAE | MSE | MAE | MSE | MAE | MSE | MAE |
| Air | 0.449 | 0.446 | 0.448 | 0.432 | 0.449 | 0.432 | 0.457 | 0.438 | 0.447 | 0.438 | 0.447 | 0.431 | 0.452 | 0.444 | 0.446 | 0.430 | **0.409** | **0.407** |
| Measles | 0.128 | 0.252 | 0.013 | 0.058 | 0.011 | 0.048 | 0.018 | 0.060 | 0.569 | 0.547 | 0.015 | 0.048 | 0.015 | 0.064 | 0.010 | 0.042 | **0.010** | **0.039** |
| SP500 | 0.630 | 0.367 | 0.523 | 0.313 | 0.516 | 0.309 | 0.611 | 0.343 | 2.674 | 1.120 | 0.511 | 0.306 | 0.568 | 0.335 | 0.555 | 0.328 | **0.441** | **0.274** |
| Atec | 0.318 | 0.314 | 0.298 | 0.298 | 0.299 | 0.275 | 0.493 | 0.429 | 0.398 | 0.387 | 0.345 | 0.319 | 0.330 | 0.339 | 0.287 | 0.280 | **0.270** | **0.266** |
| Neurolib | 1.793 | 0.381 | 2.395 | 0.438 | 2.458 | 0.445 | 2.475 | 0.458 | 2.240 | 0.532 | 1.718 | 0.347 | 2.519 | 0.451 | 1.750 | 0.350 | **1.658** | **0.339** |
| Meter | 0.944 | 0.549 | 1.254 | 0.706 | 0.941 | 0.552 | 1.034 | 0.586 | 0.987 | 0.564 | 0.949 | 0.556 | 1.308 | 0.731 | 0.943 | 0.551 | **0.933** | **0.549** |
| SIRS | 0.058 | 0.168 | 0.033 | 0.129 | 0.025 | 0.109 | 0.162 | 0.327 | 0.016 | 0.078 | 0.028 | 0.113 | 0.095 | 0.236 | 0.007 | 0.052 | **0.001** | **0.013** |
| M5 | 3.688 | 0.870 | 3.655 | 0.872 | 3.650 | 0.867 | 4.490 | 0.919 | 6.863 | 1.623 | 3.549 | 0.853 | 3.768 | 0.880 | 3.501 | 0.849 | **3.342** | **0.825** |
| Temp | 0.272 | 0.391 | 0.279 | 0.396 | 0.278 | 0.395 | 0.287 | 0.408 | 0.266 | 0.389 | 0.265 | 0.386 | 0.435 | 0.511 | 0.262 | 0.383 | **0.246** | **0.371** |
| Wind | 1.128 | 0.697 | 1.254 | 0.757 | 1.256 | 0.758 | 1.161 | 0.708 | 1.346 | 0.742 | 1.116 | 0.699 | 1.227 | 0.746 | 1.104 | 0.692 | **1.080** | **0.692** |
| Solar | 0.174 | 0.255 | 0.416 | 0.469 | 0.604 | 0.582 | 0.157 | 0.224 | 0.155 | **0.216** | 0.343 | 0.427 | — | — | 0.172 | 0.246 | **0.153** | 0.218 |
| Mobility | 0.344 | 0.359 | 0.344 | 0.341 | 0.337 | 0.336 | 0.410 | 0.388 | 1.165 | 0.787 | 0.312 | 0.314 | 0.439 | 0.410 | 0.315 | 0.317 | **0.281** | **0.292** |
| Traffic-CA | 0.063 | 0.141 | 0.295 | 0.417 | 0.491 | 0.554 | 0.101 | 0.205 | 0.082 | 0.186 | 0.271 | 0.391 | — | — | 0.061 | 0.131 | **0.054** | **0.124** |
| Wiki-20k | 10.740 | 0.394 | 10.291 | 0.305 | 10.290 | 0.306 | 10.586 | 0.325 | 10.446 | 0.332 | 10.933 | 0.405 | 10.278 | 0.304 | 10.273 | 0.302 | **10.127** | **0.290** |

memory as dimensionality increases, ConFlux remains memory-feasible across all datasets, demonstrating good scalability to thousands of variables. These results suggest that variate sorting and variate patching enable efficient handling of high-dimensional inputs by reducing unnecessary pairwise interactions while preserving essential cross-variable structure. As a result, ConFlux balances accuracy and resource usage effectively, making it well suited for large-scale multivariate forecasting settings.

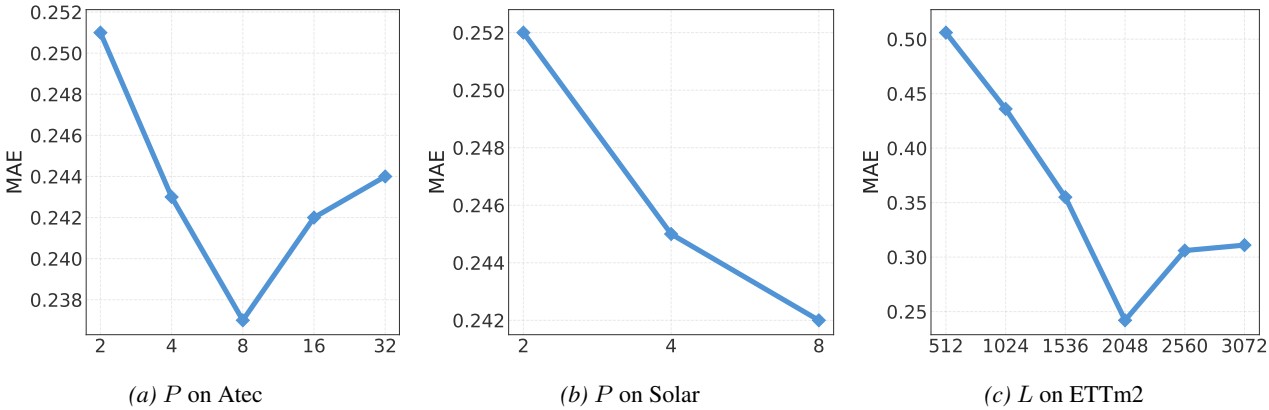

*(a) P on Atec*     *(b) P on Solar*     *(c) L on ETTm2*

*Figure 7.* The influence of patch sizes $P$ and input lengths $L$.

## J. Patch Size Impact

We investigate the effect of the variate patch size on forecasting performance to understand its role in balancing model expressivity and computational efficiency. We conduct experiments on the ETTm2 dataset, systematically varying patch size during both training and inference, and report the resulting MAE. For the training-time analysis, we evaluate ConFlux with patch sizes of 2, 4, 8, 16, and 32. As shown in Figure 7a, the MAE initially decreases as the patch size increases from 2 to 8, indicating that aggregating small groups of adjacent variables into patches can improve representation learning and reduce noise. Specifically, the lowest MAE occurs at patch size 8. When the patch size becomes larger (16 and 32), performance slightly degrades, suggesting that overly coarse patching may oversimplify variable interactions and lose fine-grained cross-variable information. We also study the impact of patch size during inference. Because the ETTm2 dataset has only 7 variables, the maximum reasonable patch size for inference is 8. We thus evaluate patch sizes of 2, 4, and 8 during inference with a model trained using the standard patch size. As shown in Figure 7b, increasing the inference patch size consistently reduces MAE, with the best performance achieved at patch size 8. This trend reinforces that moderate patches capture meaningful joint information while reducing redundant attention operations. Overall, these results demonstrate that the patch size is a crucial hyperparameter that influences both efficiency and accuracy. A moderate patch size (e.g., 16) provides a favorable trade-off: it consolidates inter-variable information for scalable attention without excessive aggregation that would obscure important details. This analysis guides practical choice of patch size for varied multivariate forecasting scenarios.

*Table 9.* Zero-shot forecasting results of time series foundation models on long-term forecasting datasets.

| Models | | ConFlux Ours | Chronos-2 2025 | | Toto 2025 | | Timer-XL 2025 | | Time-MoE 2025 | | Sundial 2025 | | TiRex 2025 | | FlowState 2025 | | Moirai 2024 | | Chronos 2024 | |
|---|---|---|---|---|---|---|---|---|---|---|---|---|---|---|---|---|---|---|---|---|---|
| Metrics | | MSE MAE | MSE | MAE | MSE | MAE | MSE | MAE | MSE | MAE | MSE | MAE | MSE | MAE | MSE | MAE | MSE | MAE | MSE | MAE |
| **ETTh1** 96 | | 0.351 0.386 | 0.352 | 0.376 | 0.396 | 0.393 | 0.369 | 0.391 | 0.357 | 0.381 | 0.348 | 0.385 | 0.367 | 0.377 | 0.376 | 0.400 | 0.376 | 0.392 | 0.440 | 0.393 |
| 192 | | 0.362 0.393 | 0.407 | 0.416 | 0.443 | 0.421 | 0.405 | 0.413 | 0.384 | 0.404 | 0.393 | 0.418 | 0.433 | 0.418 | 0.418 | 0.427 | 0.412 | 0.413 | 0.492 | 0.426 |
| 336 | | 0.377 0.406 | 0.438 | 0.439 | 0.475 | 0.436 | 0.418 | 0.423 | 0.411 | 0.434 | 0.422 | 0.440 | 0.485 | 0.447 | 0.440 | 0.441 | 0.433 | 0.428 | 0.550 | 0.462 |
| 720 | | 0.421 0.442 | 0.456 | 0.465 | 0.493 | 0.458 | 0.423 | 0.441 | 0.449 | 0.477 | 0.481 | 0.493 | 0.515 | 0.488 | 0.452 | 0.461 | 0.447 | 0.444 | 0.882 | 0.591 |
| *Avg.* | | 0.378 0.407 | 0.413 | 0.424 | 0.452 | 0.427 | 0.419 | 0.426 | 0.404 | 0.417 | 0.411 | 0.434 | 0.450 | 0.433 | 0.422 | 0.432 | 0.417 | 0.419 | 0.591 | 0.468 |
| **ETTh2** 96 | | 0.238 0.323 | 0.265 | 0.325 | 0.294 | 0.325 | 0.283 | 0.342 | 0.305 | 0.359 | 0.271 | 0.333 | 0.293 | 0.325 | 0.290 | 0.331 | 0.294 | 0.330 | 0.308 | 0.343 |
| 192 | | 0.279 0.354 | 0.337 | 0.377 | 0.361 | 0.371 | 0.340 | 0.379 | 0.351 | 0.386 | 0.327 | 0.376 | 0.355 | 0.370 | 0.357 | 0.376 | 0.365 | 0.375 | 0.384 | 0.392 |
| 336 | | 0.315 0.380 | 0.372 | 0.412 | 0.403 | 0.404 | 0.366 | 0.400 | 0.391 | 0.418 | 0.354 | 0.402 | 0.362 | 0.385 | 0.382 | 0.400 | 0.376 | 0.390 | 0.429 | 0.430 |
| 720 | | 0.404 0.438 | 0.400 | 0.443 | 0.421 | 0.430 | 0.397 | 0.431 | 0.419 | 0.454 | 0.381 | 0.435 | 0.383 | 0.416 | 0.385 | 0.413 | 0.416 | 0.433 | 0.501 | 0.477 |
| *Avg.* | | 0.309 0.374 | 0.344 | 0.389 | 0.370 | 0.383 | 0.347 | 0.388 | 0.366 | 0.404 | 0.333 | 0.387 | 0.348 | 0.374 | 0.354 | 0.380 | 0.362 | 0.382 | 0.405 | 0.410 |
| **ETTm1** 96 | | 0.268 0.329 | 0.272 | 0.313 | 0.327 | 0.341 | 0.317 | 0.356 | 0.338 | 0.368 | 0.280 | 0.334 | 0.318 | 0.329 | 0.283 | 0.340 | 0.363 | 0.356 | 0.454 | 0.408 |
| 192 | | 0.297 0.346 | 0.316 | 0.345 | 0.380 | 0.374 | 0.358 | 0.381 | 0.353 | 0.388 | 0.321 | 0.366 | 0.380 | 0.363 | 0.329 | 0.368 | 0.388 | 0.375 | 0.567 | 0.477 |
| 336 | | 0.321 0.363 | 0.346 | 0.369 | 0.414 | 0.398 | 0.386 | 0.401 | 0.381 | 0.413 | 0.350 | 0.389 | 0.414 | 0.386 | 0.361 | 0.390 | 0.416 | 0.392 | 0.662 | 0.525 |
| 720 | | 0.371 0.395 | 0.412 | 0.414 | 0.489 | 0.439 | 0.430 | 0.431 | 0.504 | 0.493 | 0.394 | 0.418 | 0.477 | 0.420 | 0.421 | 0.424 | 0.460 | 0.418 | 0.900 | 0.591 |
| *Avg.* | | 0.314 0.358 | 0.337 | 0.360 | 0.403 | 0.388 | 0.373 | 0.392 | 0.394 | 0.415 | 0.336 | 0.377 | 0.397 | 0.374 | 0.349 | 0.380 | 0.406 | 0.385 | 0.645 | 0.500 |
| **ETTm2** 96 | | 0.168 0.255 | 0.151 | 0.238 | 0.178 | 0.244 | 0.189 | 0.277 | 0.201 | 0.291 | 0.170 | 0.256 | 0.178 | 0.243 | 0.160 | 0.239 | 0.205 | 0.273 | 0.199 | 0.274 |
| 192 | | 0.221 0.291 | 0.211 | 0.284 | 0.241 | 0.290 | 0.241 | 0.315 | 0.258 | 0.334 | 0.229 | 0.300 | 0.257 | 0.298 | 0.224 | 0.285 | 0.275 | 0.316 | 0.261 | 0.322 |
| 336 | | 0.259 0.318 | 0.266 | 0.324 | 0.300 | 0.331 | 0.286 | 0.348 | 0.324 | 0.373 | 0.281 | 0.337 | 0.325 | 0.344 | 0.279 | 0.324 | 0.329 | 0.350 | 0.326 | 0.366 |
| 720 | | 0.322 0.361 | 0.353 | 0.382 | 0.384 | 0.386 | 0.375 | 0.402 | 0.488 | 0.464 | 0.351 | 0.387 | 0.400 | 0.398 | 0.362 | 0.384 | 0.437 | 0.411 | 0.455 | 0.439 |
| *Avg.* | | 0.242 0.306 | 0.245 | 0.307 | 0.276 | 0.313 | 0.273 | 0.336 | 0.317 | 0.365 | 0.258 | 0.320 | 0.290 | 0.321 | 0.256 | 0.308 | 0.311 | 0.337 | 0.310 | 0.350 |
| **Weather** 96 | | 0.172 0.219 | 0.150 | 0.192 | 0.161 | 0.201 | 0.171 | 0.225 | 0.160 | 0.214 | 0.157 | 0.205 | 0.161 | 0.191 | 0.184 | 0.227 | 0.220 | 0.217 | 0.194 | 0.235 |
| 192 | | 0.209 0.249 | 0.213 | 0.252 | 0.210 | 0.249 | 0.221 | 0.271 | 0.210 | 0.260 | 0.205 | 0.251 | 0.224 | 0.251 | 0.228 | 0.274 | 0.271 | 0.259 | 0.249 | 0.285 |
| 336 | | 0.243 0.274 | 0.292 | 0.307 | 0.270 | 0.294 | 0.274 | 0.311 | 0.274 | 0.309 | 0.253 | 0.289 | 0.294 | 0.301 | 0.275 | 0.313 | 0.286 | 0.297 | 0.302 | 0.327 |
| 720 | | 0.294 0.309 | 0.422 | 0.378 | 0.349 | 0.348 | 0.356 | 0.370 | 0.418 | 0.405 | 0.320 | 0.336 | 0.395 | 0.362 | 0.346 | 0.366 | 0.373 | 0.354 | 0.372 | 0.378 |
| *Avg.* | | 0.229 0.263 | 0.269 | 0.282 | 0.248 | 0.273 | 0.256 | 0.294 | 0.265 | 0.297 | 0.234 | 0.270 | 0.269 | 0.276 | 0.258 | 0.295 | 0.281 | 0.287 | 0.279 | 0.306 |
| **Solar** 96 | | 0.144 0.196 | 0.143 | 0.184 | 0.288 | 0.238 | 0.591 | 0.504 | 0.304 | 0.345 | 0.187 | 0.213 | 0.194 | 0.196 | 0.706 | 0.654 | 0.682 | 0.688 | 0.373 | 0.304 |
| 192 | | 0.157 0.206 | 0.163 | 0.201 | 0.330 | 0.258 | 0.689 | 0.567 | 0.309 | 0.342 | 0.210 | 0.235 | 0.222 | 0.213 | 0.760 | 0.727 | 0.694 | 0.695 | 0.363 | 0.303 |
| 336 | | 0.170 0.214 | 0.179 | 0.214 | 0.389 | 0.276 | 0.831 | 0.636 | 0.433 | 0.450 | 0.231 | 0.254 | 0.240 | 0.224 | 0.750 | 0.746 | 0.719 | 0.706 | 0.391 | 0.319 |
| 720 | | 0.179 0.221 | 0.195 | 0.224 | 0.962 | 0.312 | 0.972 | 0.710 | 0.599 | 0.576 | 0.254 | 0.276 | 0.232 | 0.221 | 0.768 | 0.781 | 0.759 | 0.725 | 0.444 | 0.349 |
| *Avg.* | | 0.162 0.209 | 0.170 | 0.206 | 0.492 | 0.271 | 0.771 | 0.604 | 0.411 | 0.428 | 0.221 | 0.245 | 0.222 | 0.214 | 0.746 | 0.727 | 0.714 | 0.704 | 0.393 | 0.319 |
| **Electricity** 96 | | 0.145 0.251 | 0.123 | 0.214 | 0.146 | 0.235 | 0.141 | 0.237 | - | - | 0.132 | 0.229 | 0.204 | 0.265 | 0.190 | 0.289 | 0.160 | 0.250 | 0.154 | 0.231 |
| 192 | | 0.157 0.261 | 0.142 | 0.233 | 0.165 | 0.253 | 0.159 | 0.254 | - | - | 0.152 | 0.250 | 0.265 | 0.307 | 0.202 | 0.300 | 0.175 | 0.263 | 0.179 | 0.254 |
| 336 | | 0.167 0.271 | 0.182 | 0.254 | 0.188 | 0.274 | 0.177 | 0.272 | - | - | 0.173 | 0.271 | 0.297 | 0.332 | 0.220 | 0.317 | 0.187 | 0.277 | 0.214 | 0.284 |
| 720 | | 0.190 0.291 | 0.672 | 0.292 | 0.255 | 0.321 | 0.219 | 0.308 | - | - | 0.218 | 0.311 | 0.310 | 0.349 | 0.261 | 0.350 | 0.228 | 0.309 | 0.311 | 0.346 |
| *Avg.* | | 0.165 0.269 | 0.280 | 0.248 | 0.189 | 0.271 | 0.174 | 0.278 | - | - | 0.169 | 0.265 | 0.269 | 0.313 | 0.218 | 0.314 | 0.187 | 0.274 | 0.214 | 0.278 |

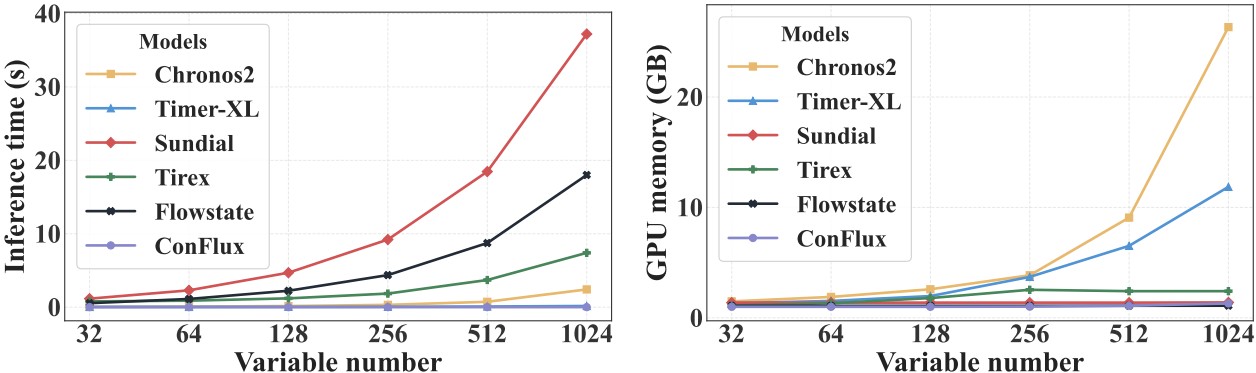

*Figure 8.* Inference efficiency comparison.

## K. Lookback Window Length Impact

To understand how the length of historical context affects forecasting performance, we evaluate ConFlux under varying lookback window lengths at inference time while keeping the training lookback fixed at 2048. Because our ConFlux can employ a Fourier basis representation prior to variate embedding, it can naturally accept inputs of differing lengths by mapping time series segments onto a unified functional basis. This design enables flexible inference lengths without retraining. We conduct experiments on the ETTm2 dataset using lookback lengths of 512, 1024, 1536, 2048, 2560, and 3072. Figure 7c presents the resulting MAE values for each setting. As the lookback length increases from 512 to 2048, forecasting performance improves markedly, demonstrating that incorporating more historical context allows the model to capture richer temporal dependencies and better inform predictions. The best performance is achieved at a lookback length of 2048, which matches the training context length. When the lookback length is extended beyond the training length, performance deteriorates relative to the optimal 2048 input. This suggests that, although ConFlux's Fourier basis mechanism enables different inference lengths, including excessively distant history may introduce less relevant or noisy information, which can outweigh the benefits of additional context for the forecasting task. These results highlight two key properties of ConFlux. First, the model reliably exploits increasing historical context up to the training length, benefiting from longer lookbacks. Second, the Fourier basis representation provides flexibility in input length at inference time, a desirable characteristic for real-world applications where available context may vary.

## L. Inference Efficiency

To evaluate scalability and practical applicability of ConFlux, we measure inference efficiency across varying numbers of variables between our ConFlux and several representative time series forecasting models that embody different architectural paradigm in Figure 8.

- Chronos-2: a multivariate foundation model which stacks time and variate attention.
- Timer-XL: a multivariate foundation model which flattens variate and temporal dimensions.
- Sundial: a Transformer-based univariate foundation model.
- TiRex: a LSTM-based univariate foundation model.
- FlowState: a state space model (SSM)-based univariate foundation model.

Specifically, we measure peak GPU memory usage (in MB) and inference latency (in microseconds) under variable counts: 32, 64, 128, 256, 512, and 1024. Lower values for both memory and time indicate better efficiency. As summarized in Figure 5, ConFlux consistently demonstrates superior inference efficiency across all tested variable dimensions. When variables are small (32 and 64), ConFlux uses only modest memory while maintaining very low latency, substantially outperforming models with explicit variate attention such as Chronos-2 and Timer-XL. As variable count increases, the efficiency gap becomes more pronounced. For example, at 1024 variables, ConFlux's memory usage remains below 1400 MB and latency under 6 μs, whereas Chronos-2's memory consumption increases dramatically and latency grows by over two orders of magnitude, reflecting the cost of full variate attention. Timer-XL, which flattens variate and temporal axes, also exhibits significant growth in both memory and latency as variable count increases. Univariate models (Sundial, TiRex,

FlowState) behave differently: because they process each variable independently, their memory footprint remains relatively stable across dimensions, but inference latency increases substantially with variable count due to repeated per-variable computation. In contrast, ConFlux's variate patching strategy effectively limits both memory and computational growth, enabling efficient joint inference across many variables. These results highlight the practical advantages of ConFlux for multivariate time series forecasting: it maintains competitive accuracy while delivering scalable inference efficiency. Moreover, ConFlux scales gracefully in both memory usage and latency as variable count increases, making it well suited for applications with large numbers of series where traditional attention-based approaches become computationally prohibitive.

*Table 10.* Zero-shot forecasting results of time series foundation models on short-term forecasting datasets.

| Models | ConFlux (Ours) | | Chronos-2 (2025) | | FlowState (2025) | | Chronos (2024) | |
|---|---|---|---|---|---|---|---|---|
| Metrics | MSE | MAE | MSE | MAE | MSE | MAE | MSE | MAE |
| **PEMS03** 12 | 0.065 | 0.172 | **0.061** | **0.160** | 0.105 | 0.219 | 0.074 | 0.179 |
| 24 | **0.075** | **0.182** | 0.086 | 0.187 | 0.191 | 0.296 | 0.106 | 0.207 |
| 48 | **0.091** | **0.196** | 0.151 | 0.237 | 0.393 | 0.429 | 0.159 | 0.247 |
| 96 | **0.107** | **0.208** | 0.230 | 0.291 | 0.784 | 0.635 | 0.207 | 0.281 |
| *Avg.* | **0.085** | **0.190** | 0.132 | 0.219 | 0.368 | 0.395 | 0.136 | 0.229 |
| **PEMS04** 12 | **0.074** | 0.181 | 0.074 | **0.176** | 0.116 | 0.229 | 0.090 | 0.199 |
| 24 | **0.083** | **0.191** | 0.096 | 0.202 | 0.197 | 0.303 | 0.125 | 0.232 |
| 48 | **0.095** | **0.203** | 0.152 | 0.252 | 0.398 | 0.437 | 0.184 | 0.279 |
| 96 | **0.106** | **0.210** | 0.226 | 0.307 | 0.790 | 0.647 | 0.239 | 0.318 |
| *Avg.* | **0.090** | **0.196** | 0.137 | 0.234 | 0.375 | 0.404 | 0.160 | 0.257 |
| **PEMS07** 12 | 0.059 | 0.160 | **0.056** | **0.154** | 0.092 | 0.206 | 0.073 | 0.172 |
| 24 | **0.066** | **0.168** | 0.077 | 0.181 | 0.164 | 0.278 | 0.106 | 0.211 |
| 48 | **0.076** | **0.178** | 0.131 | 0.232 | 0.343 | 0.407 | 0.187 | 0.256 |
| 96 | **0.083** | **0.184** | 0.205 | 0.290 | 0.682 | 0.605 | 0.322 | 0.295 |
| *Avg.* | **0.071** | **0.173** | 0.117 | 0.214 | 0.320 | 0.374 | 0.172 | 0.233 |
| **PEMS08** 12 | 0.083 | 0.179 | **0.068** | **0.170** | 0.107 | 0.220 | 0.082 | 0.190 |
| 24 | 0.103 | **0.190** | **0.095** | 0.200 | 0.183 | 0.290 | 0.120 | 0.225 |
| 48 | **0.142** | **0.204** | 0.165 | 0.258 | 0.375 | 0.424 | 0.194 | 0.279 |
| 96 | **0.200** | **0.215** | 0.282 | 0.332 | 0.813 | 0.650 | 0.297 | 0.334 |
| *Avg.* | **0.132** | **0.197** | 0.152 | 0.240 | 0.370 | 0.396 | 0.173 | 0.257 |

## M. Forecasting without In-domain Fine-tuning — Full Results

In this section, we provide the full set of zero-shot forecasting results across multiple long-term and short-term benchmarks. Table 9 and Table 10 report detailed performance for each forecast horizon and dataset.

Several patterns emerge from these comprehensive results. First, ConFlux consistently achieves the best or near-best performance across nearly all forecasting tasks without any in-domain fine-tuning. This trend holds across a wide variety of domains, forecast lengths, and evaluation metrics, underscoring the generality of the learned multivariate representations. Second, multivariate foundation models in general, including ConFlux and Chronos-2, obtain relatively larger improvements on datasets with higher variable dimensionality, such as the PEMS traffic datasets. In these high-dimensional scenarios, the ability to jointly reason over many channels confers a clear advantage over single-variable models. Single-variable approaches such as FlowState and Chronos tend to lag behind in high-dimensional settings, as they must independently model each channel without capturing cross-variable dependencies. Third, the advantage of ConFlux is particularly pronounced in long-term forecasting. On long horizons, ConFlux frequently outperforms baselines by a substantial margin. We attribute this to the non-autoregressive Fourier basis decoder, which directly synthesizes future values by reconstructing from learned frequency components, rather than relying on iterative step-by-step prediction. This design helps preserve global temporal structure and reduces error accumulation over long intervals, resulting in stronger zero-shot performance at extended horizons. Together, these results demonstrate the robustness and versatility of ConFlux as a zero-shot multivariate foundation model.

## N. Forecasting with In-domain Fine-tuning — Full Results

In addition to zero-shot evaluation, we also investigate the performance of models when fine-tuned on in-domain training data. Tables 11 report the detailed forecasting results after fine-tuning.

From these full results, several consistent patterns emerge. First, fine-tuning leads to substantial performance gains for ConFlux compared to its zero-shot counterparts. Across datasets and forecast horizons, ConFlux's MAE and MSE decrease markedly after in-domain adaptation, demonstrating that the learned foundation representations are amenable to domain-specific refinement. This improvement reflects ConFlux's strong generalization capability: the pre-trained model provides a robust initialization that is highly adaptable to domain characteristics when fine-tuned, enabling better alignment with local temporal dynamics and variable interactions present in the target dataset. Second, we observe that certain single-variable baselines, such as Time-MoE and Chronos, narrow the performance gap or even outperform some multivariate baselines like Timer-XL after fine-tuning. A plausible explanation is that, during fine-tuning, exposure to in-domain temporal patterns allows these models — which specialize in capturing strong per-series temporal dynamics — to leverage domain-specific time dependencies more effectively.

## O. Forecasting with From-scratch Training — Full Results

In this section, we present the complete results of multivariate forecasting models trained from scratch on in-domain training data. Tables 12, 13 and 8 summarize performance across long-term, short-term, and high-dimensional datasets.

Several consistent patterns can be observed from these full results. First, ConFlux achieves the best overall performance across nearly all datasets, forecasting horizons, and domains. This includes long-term benchmarks such as the ETT series, short-term traffic datasets, and high-dimensional scenarios. The superior performance of ConFlux in from-scratch training demonstrates not only strong generalization ability but also excellent task-specific modeling capability when trained directly on in-domain data. Second, the gains achieved by ConFlux are especially marked in high-dimensional settings. On these tasks, the performance improvement relative to competing methods is more pronounced, providing evidence that our choice to include high-dimensional series in the training corpus was instrumental. Third, on high-variable tasks such as the PEMS datasets, multivariate models like DUET outperform single-variable baselines, highlighting the advantage of capturing cross-channel interactions. Together, these observations demonstrate that Conflux not only generalizes well across domains but also excels when trained from scratch within a specific domain.

## P. Broad Impact

**Real-world applications**   ConFlux is designed as a general-purpose foundation model for multivariate time series forecasting, with strong performance in zero-shot, fine-tuning, and from-scratch settings across a wide range of benchmarks. Its ability to model large-scale, high-dimensional multivariate systems with low memory cost and fast inference makes it well suited for real-world scenarios where many variables interact asynchronously. Typical applications include energy systems with many sensors and strong cross-variable dependence, traffic and transportation networks with dense spatial correlations, climate and weather forecasting involving large numbers of atmospheric variables, and industrial monitoring systems with heterogeneous signals. By sorting variables based on data-driven associations and aggregating related variables into patches, ConFlux can handle noise, scale to thousands of variables, and maintain stable forecasting accuracy, which is critical for deployment in complex operational environments.

**Academic research**   From a research perspective, ConFlux provides a new modeling paradigm for multivariate time series forecasting by shifting the focus from temporal tokenization to variate-level organization. The proposed variate sorting and variate patching mechanisms offer a principled way to reduce attention complexity while preserving cross-variable structure, which addresses a key limitation of existing multivariate transformers. In addition, the use of a Fourier basis decoder enables flexible forecasting over arbitrary horizons without architectural changes, offering an alternative to fixed-length decoders commonly used in prior work. These ideas open several research directions, including scalable multivariate foundation models, structure-aware variable organization, and hybrid time–frequency decoding within deep forecasting systems.

**Model robustness and scalability**   The robustness of ConFlux is supported by extensive evaluations on diverse datasets with varying numbers of variables, time scales, and dependency patterns. Experimental results show that ConFlux maintains strong accuracy under distribution shift in zero-shot settings, while also benefiting from in-domain fine-tuning when data are available. Its inference cost grows smoothly with the number of variables, making it suitable for large-scale multivariate forecasting tasks that are difficult for standard attention-based models. This combination of robustness, scalability, and efficiency suggests that ConFlux can serve as a reliable foundation for future multivariate time series applications and further research on general forecasting models.

*Table 11.* Fine-tune forecasting results of time series foundation models on long-term forecasting datasets.

| Models | ConFlux Ours | | Timer-XL 2025 | | Time-MoE 2025 | | Moirai 2024 | | Chronos 2024 | |
|---|---|---|---|---|---|---|---|---|---|---|
| Metrics | MSE | MAE | MSE | MAE | MSE | MAE | MSE | MAE | MSE | MAE |
| **ETTh1 96** | 0.342 | 0.379 | 0.365 | 0.393 | **0.340** | 0.377 | 0.367 | 0.396 | 0.366 | **0.374** |
| 192 | **0.349** | **0.384** | 0.400 | 0.416 | 0.380 | 0.408 | 0.405 | 0.418 | 0.418 | 0.405 |
| 336 | **0.360** | **0.395** | 0.419 | 0.431 | 0.406 | 0.432 | 0.435 | 0.434 | 0.453 | 0.424 |
| 720 | **0.399** | **0.432** | 0.454 | 0.465 | 0.437 | 0.469 | 0.465 | 0.459 | 0.452 | 0.441 |
| *Avg.* | **0.363** | **0.397** | 0.410 | 0.426 | 0.391 | 0.422 | 0.418 | 0.427 | 0.422 | 0.411 |
| **ETTh2 96** | **0.235** | 0.319 | 0.287 | 0.345 | 0.267 | 0.330 | 0.273 | 0.338 | 0.266 | **0.311** |
| 192 | **0.277** | **0.352** | 0.344 | 0.385 | 0.341 | 0.380 | 0.331 | 0.385 | 0.334 | 0.357 |
| 336 | **0.312** | **0.378** | 0.375 | 0.410 | 0.386 | 0.415 | 0.358 | 0.410 | 0.378 | 0.391 |
| 720 | 0.398 | 0.436 | 0.411 | 0.445 | 0.419 | 0.454 | **0.392** | 0.444 | 0.395 | **0.418** |
| *Avg.* | **0.305** | 0.371 | 0.354 | 0.396 | 0.353 | 0.395 | 0.339 | 0.394 | 0.343 | **0.369** |
| **ETTm1 96** | 0.253 | 0.320 | 0.305 | 0.355 | **0.220** | 0.302 | 0.309 | 0.343 | 0.285 | **0.315** |
| 192 | **0.282** | **0.339** | 0.348 | 0.383 | 0.284 | 0.347 | 0.349 | 0.367 | 0.329 | 0.345 |
| 336 | **0.305** | **0.355** | 0.382 | 0.406 | 0.349 | 0.392 | 0.379 | 0.385 | 0.361 | 0.369 |
| 720 | **0.351** | **0.386** | 0.428 | 0.439 | 0.462 | 0.466 | 0.410 | 0.409 | 0.434 | 0.413 |
| *Avg.* | **0.298** | **0.350** | 0.366 | 0.396 | 0.329 | 0.377 | 0.362 | 0.376 | 0.352 | 0.361 |
| **ETTm2 96** | **0.143** | 0.238 | 0.191 | 0.280 | 0.161 | 0.253 | 0.174 | 0.261 | 0.160 | **0.231** |
| 192 | **0.186** | **0.269** | 0.245 | 0.320 | 0.229 | 0.308 | 0.234 | 0.304 | 0.225 | 0.276 |
| 336 | **0.219** | **0.296** | 0.297 | 0.356 | 0.313 | 0.367 | 0.284 | 0.340 | 0.286 | 0.317 |
| 720 | **0.273** | **0.339** | 0.389 | 0.416 | 0.486 | 0.464 | 0.353 | 0.386 | 0.381 | 0.377 |
| *Avg.* | **0.205** | **0.285** | 0.281 | 0.343 | 0.297 | 0.348 | 0.261 | 0.323 | 0.263 | 0.300 |
| **Weather 96** | 0.151 | 0.197 | 0.169 | 0.226 | **0.141** | **0.194** | 0.178 | 0.228 | 0.174 | 0.209 |
| 192 | **0.185** | **0.231** | 0.217 | 0.271 | 0.196 | 0.254 | 0.220 | 0.269 | 0.231 | 0.258 |
| 336 | **0.219** | **0.260** | 0.267 | 0.308 | 0.260 | 0.309 | 0.308 | 0.330 | 0.289 | 0.298 |
| 720 | **0.268** | **0.298** | 0.336 | 0.358 | 0.346 | 0.378 | 0.390 | 0.388 | 0.376 | 0.347 |
| *Avg.* | **0.206** | **0.246** | 0.247 | 0.291 | 0.236 | 0.284 | 0.274 | 0.304 | 0.268 | 0.278 |
| **Solar 96** | **0.131** | **0.224** | 0.523 | 0.461 | 0.375 | 0.320 | 0.412 | 0.431 | 0.174 | 0.276 |
| 192 | **0.143** | **0.196** | 0.556 | 0.482 | 0.393 | 0.331 | 0.425 | 0.444 | 0.179 | 0.280 |
| 336 | **0.155** | **0.204** | 0.612 | 0.534 | 0.428 | 0.345 | 0.457 | 0.468 | 0.188 | 0.291 |
| 720 | **0.163** | **0.211** | 0.714 | 0.618 | 0.496 | 0.378 | 0.506 | 0.490 | 0.205 | 0.307 |
| *Avg.* | **0.148** | **0.199** | 0.601 | 0.524 | 0.423 | 0.344 | 0.450 | 0.458 | 0.187 | 0.288 |
| **Electricity 96** | 0.131 | 0.224 | 0.139 | 0.237 | - | - | 0.152 | 0.247 | **0.121** | **0.208** |
| 192 | 0.150 | 0.242 | 0.158 | 0.256 | - | - | 0.170 | 0.263 | **0.139** | **0.226** |
| 336 | 0.166 | 0.260 | 0.177 | 0.275 | - | - | 0.192 | 0.285 | **0.157** | **0.244** |
| 720 | **0.188** | 0.287 | 0.219 | 0.313 | - | - | 0.220 | 0.307 | 0.199 | **0.283** |
| *Avg.* | 0.159 | 0.253 | 0.173 | 0.270 | - | - | 0.184 | 0.276 | **0.154** | **0.240** |

Table 12. Results of long-term from-scratch forecasting. All results are reported on four forecasting horizon: $H \in \{96, 192, 336, 720\}$. The best and second-best results are highlighted in **bold** and underlined, respectively.

| Models | | ConFlux (Ours) | | DUET 2025 | | CycleNet 2024 | | TimeXer 2024 | | SOFTS 2024 | | iTransformer 2024 | | PatchTST 2024 | | Crossformer 2023 | | TimesNet 2023 | | DLinear 2023 | | SCINet 2022 | |
|---|---|---|---|---|---|---|---|---|---|---|---|---|---|---|---|---|---|---|---|---|---|---|---|
| Metrics | | MSE | MAE | MSE | MAE | MSE | MAE | MSE | MAE | MSE | MAE | MSE | MAE | MSE | MAE | MSE | MAE | MSE | MAE | MSE | MAE | MSE | MAE |
| ETTh1 | 96 | **0.371** | **0.388** | 0.377 | 0.393 | 0.375 | 0.395 | 0.382 | 0.403 | 0.381 | 0.399 | 0.386 | 0.405 | 0.414 | 0.419 | 0.423 | 0.448 | 0.384 | 0.402 | 0.386 | 0.400 | 0.654 | 0.599 |
| | 192 | **0.424** | **0.422** | 0.429 | 0.425 | 0.436 | 0.428 | 0.429 | 0.435 | 0.435 | 0.431 | 0.441 | 0.436 | 0.460 | 0.445 | 0.471 | 0.474 | 0.436 | 0.429 | 0.437 | 0.432 | 0.719 | 0.631 |
| | 336 | **0.465** | **0.442** | 0.471 | 0.446 | 0.496 | 0.455 | 0.468 | 0.448 | 0.480 | 0.452 | 0.487 | 0.458 | 0.501 | 0.466 | 0.570 | 0.546 | 0.491 | 0.469 | 0.481 | 0.459 | 0.778 | 0.659 |
| | 720 | **0.461** | **0.461** | 0.496 | 0.480 | 0.520 | 0.484 | 0.469 | 0.461 | 0.499 | 0.488 | 0.503 | 0.491 | 0.500 | 0.488 | 0.653 | 0.621 | 0.521 | 0.500 | 0.519 | 0.516 | 0.836 | 0.699 |
| | *Avg.* | **0.430** | **0.428** | 0.443 | 0.436 | 0.457 | 0.441 | 0.437 | 0.437 | 0.449 | 0.442 | 0.454 | 0.448 | 0.469 | 0.455 | 0.529 | 0.522 | 0.458 | 0.450 | 0.456 | 0.452 | 0.747 | 0.647 |
| ETTh2 | 96 | **0.284** | **0.335** | 0.296 | 0.345 | 0.298 | 0.344 | 0.286 | 0.338 | 0.297 | 0.347 | 0.297 | 0.349 | 0.302 | 0.348 | 0.745 | 0.584 | 0.340 | 0.374 | 0.333 | 0.387 | 0.707 | 0.621 |
| | 192 | **0.362** | **0.387** | 0.368 | 0.389 | 0.372 | 0.396 | 0.363 | 0.389 | 0.373 | 0.394 | 0.380 | 0.400 | 0.388 | 0.400 | 0.877 | 0.656 | 0.402 | 0.414 | 0.477 | 0.476 | 0.860 | 0.689 |
| | 336 | **0.404** | **0.414** | 0.411 | 0.422 | 0.431 | 0.439 | 0.414 | 0.423 | 0.410 | 0.426 | 0.428 | 0.432 | 0.426 | 0.433 | 1.043 | 0.731 | 0.452 | 0.452 | 0.594 | 0.541 | 1.000 | 0.744 |
| | 720 | **0.400** | 0.428 | 0.412 | **0.424** | 0.450 | 0.458 | 0.408 | 0.432 | 0.411 | 0.433 | 0.427 | 0.445 | 0.431 | 0.446 | 1.104 | 0.763 | 0.462 | 0.468 | 0.831 | 0.657 | 1.249 | 0.838 |
| | *Avg.* | **0.363** | **0.391** | 0.372 | 0.397 | 0.388 | 0.409 | 0.367 | 0.396 | 0.385 | 0.408 | 0.383 | 0.407 | 0.387 | 0.407 | 0.942 | 0.684 | 0.414 | 0.427 | 0.559 | 0.515 | 0.954 | 0.723 |
| ETTm1 | 96 | **0.312** | **0.340** | 0.324 | 0.354 | 0.325 | 0.363 | 0.318 | 0.356 | 0.325 | 0.361 | 0.334 | 0.368 | 0.329 | 0.367 | 0.404 | 0.426 | 0.338 | 0.375 | 0.345 | 0.372 | 0.418 | 0.438 |
| | 192 | **0.358** | **0.372** | 0.369 | 0.379 | 0.366 | 0.382 | 0.362 | 0.383 | 0.375 | 0.389 | 0.377 | 0.391 | 0.367 | 0.385 | 0.450 | 0.451 | 0.374 | 0.387 | 0.380 | 0.389 | 0.439 | 0.450 |
| | 336 | **0.389** | **0.394** | 0.404 | 0.402 | 0.396 | 0.401 | 0.395 | 0.407 | 0.405 | 0.412 | 0.426 | 0.420 | 0.399 | 0.410 | 0.532 | 0.515 | 0.410 | 0.411 | 0.413 | 0.413 | 0.490 | 0.485 |
| | 720 | **0.457** | **0.433** | 0.463 | 0.437 | 0.457 | 0.433 | 0.452 | 0.441 | 0.466 | 0.447 | 0.491 | 0.459 | 0.454 | 0.439 | 0.666 | 0.589 | 0.478 | 0.450 | 0.474 | 0.453 | 0.595 | 0.550 |
| | *Avg.* | **0.379** | **0.385** | 0.390 | 0.393 | 0.386 | 0.395 | 0.382 | 0.397 | 0.393 | 0.403 | 0.407 | 0.410 | 0.387 | 0.400 | 0.513 | 0.495 | 0.400 | 0.406 | 0.403 | 0.407 | 0.486 | 0.481 |
| ETTm2 | 96 | **0.166** | **0.247** | 0.174 | 0.255 | 0.166 | 0.248 | 0.182 | 0.266 | 0.180 | 0.261 | 0.180 | 0.264 | 0.175 | 0.259 | 0.287 | 0.366 | 0.187 | 0.267 | 0.193 | 0.292 | 0.286 | 0.377 |
| | 192 | **0.233** | 0.293 | 0.243 | 0.302 | 0.233 | **0.291** | 0.248 | 0.306 | 0.246 | 0.306 | 0.250 | 0.309 | 0.241 | 0.302 | 0.414 | 0.492 | 0.249 | 0.309 | 0.284 | 0.362 | 0.399 | 0.445 |
| | 336 | **0.293** | 0.334 | 0.304 | 0.341 | 0.293 | **0.330** | 0.312 | 0.346 | 0.319 | 0.352 | 0.311 | 0.348 | 0.305 | 0.343 | 0.597 | 0.542 | 0.321 | 0.351 | 0.369 | 0.427 | 0.637 | 0.591 |
| | 720 | **0.395** | 0.392 | 0.399 | 0.397 | 0.395 | **0.389** | 0.414 | 0.404 | 0.405 | 0.401 | 0.412 | 0.407 | 0.402 | 0.400 | 1.730 | 1.042 | 0.408 | 0.403 | 0.554 | 0.522 | 0.960 | 0.735 |
| | *Avg.* | **0.272** | 0.317 | 0.280 | 0.324 | 0.272 | **0.315** | 0.289 | 0.330 | 0.287 | 0.330 | 0.288 | 0.332 | 0.281 | 0.326 | 0.757 | 0.611 | 0.291 | 0.333 | 0.350 | 0.401 | 0.571 | 0.537 |
| Weather | 96 | **0.153** | **0.192** | 0.163 | 0.202 | 0.158 | 0.203 | 0.157 | 0.205 | 0.166 | 0.208 | 0.174 | 0.214 | 0.177 | 0.210 | 0.158 | 0.230 | 0.172 | 0.220 | 0.196 | 0.255 | 0.221 | 0.306 |
| | 192 | **0.203** | **0.239** | 0.218 | 0.252 | 0.207 | 0.247 | 0.204 | 0.247 | 0.217 | 0.253 | 0.221 | 0.254 | 0.225 | 0.250 | 0.206 | 0.277 | 0.219 | 0.261 | 0.237 | 0.296 | 0.261 | 0.340 |
| | 336 | **0.260** | **0.282** | 0.274 | 0.294 | 0.262 | 0.289 | 0.261 | 0.290 | 0.282 | 0.300 | 0.278 | 0.296 | 0.278 | 0.290 | 0.272 | 0.335 | 0.280 | 0.306 | 0.283 | 0.335 | 0.309 | 0.378 |
| | 720 | 0.341 | **0.334** | 0.349 | 0.343 | 0.344 | 0.344 | **0.340** | 0.341 | 0.356 | 0.351 | 0.358 | 0.349 | 0.354 | 0.340 | 0.398 | 0.418 | 0.365 | 0.359 | 0.345 | 0.381 | 0.377 | 0.427 |
| | *Avg.* | **0.239** | **0.262** | 0.251 | 0.273 | 0.243 | 0.271 | 0.241 | 0.271 | 0.255 | 0.278 | 0.258 | 0.278 | 0.259 | 0.273 | 0.259 | 0.315 | 0.259 | 0.287 | 0.265 | 0.317 | 0.292 | 0.363 |
| Solar | 96 | **0.181** | 0.218 | 0.200 | **0.207** | 0.190 | 0.247 | 0.215 | 0.295 | 0.200 | 0.230 | 0.203 | 0.237 | 0.205 | 0.246 | 0.310 | 0.331 | 0.250 | 0.292 | 0.290 | 0.378 | 0.237 | 0.344 |
| | 192 | **0.205** | 0.241 | 0.228 | **0.233** | 0.210 | 0.266 | 0.236 | 0.301 | 0.229 | 0.253 | 0.233 | 0.261 | 0.237 | 0.267 | 0.734 | 0.725 | 0.296 | 0.318 | 0.320 | 0.398 | 0.280 | 0.380 |
| | 336 | 0.218 | 0.250 | 0.262 | **0.244** | 0.217 | 0.266 | 0.252 | 0.307 | 0.243 | 0.269 | 0.248 | 0.273 | 0.250 | 0.276 | 0.750 | 0.735 | 0.319 | 0.330 | 0.353 | 0.415 | 0.304 | 0.389 |
| | 720 | **0.222** | 0.257 | 0.258 | **0.249** | 0.223 | 0.266 | 0.244 | 0.305 | 0.245 | 0.272 | 0.249 | 0.275 | 0.252 | 0.275 | 0.769 | 0.765 | 0.338 | 0.337 | 0.356 | 0.413 | 0.308 | 0.388 |
| | *Avg.* | **0.207** | 0.242 | 0.237 | **0.233** | 0.210 | 0.261 | 0.237 | 0.302 | 0.229 | 0.256 | 0.233 | 0.262 | 0.236 | 0.266 | 0.641 | 0.639 | 0.301 | 0.319 | 0.330 | 0.401 | 0.282 | 0.375 |
| Electricity | 96 | **0.132** | **0.224** | 0.145 | 0.233 | 0.136 | 0.229 | 0.140 | 0.242 | 0.143 | 0.233 | 0.148 | 0.240 | 0.181 | 0.270 | 0.219 | 0.314 | 0.168 | 0.272 | 0.197 | 0.282 | 0.286 | 0.377 |
| | 192 | **0.150** | **0.240** | 0.163 | 0.248 | 0.152 | 0.244 | 0.157 | 0.256 | 0.158 | 0.248 | 0.162 | 0.253 | 0.188 | 0.274 | 0.231 | 0.322 | 0.184 | 0.289 | 0.196 | 0.285 | 0.399 | 0.445 |
| | 336 | **0.163** | **0.256** | 0.175 | 0.262 | 0.170 | 0.264 | 0.176 | 0.275 | 0.178 | 0.269 | 0.178 | 0.269 | 0.204 | 0.293 | 0.246 | 0.337 | 0.198 | 0.300 | 0.209 | 0.301 | 0.637 | 0.591 |
| | 720 | **0.190** | **0.281** | 0.204 | 0.291 | 0.212 | 0.299 | 0.211 | 0.306 | 0.218 | 0.305 | 0.225 | 0.317 | 0.246 | 0.324 | 0.280 | 0.363 | 0.220 | 0.320 | 0.245 | 0.333 | 0.960 | 0.735 |
| | *Avg.* | **0.159** | **0.250** | 0.172 | 0.258 | 0.168 | 0.259 | 0.171 | 0.270 | 0.174 | 0.264 | 0.178 | 0.270 | 0.205 | 0.290 | 0.244 | 0.334 | 0.193 | 0.295 | 0.212 | 0.300 | 0.571 | 0.537 |

*Table 13.* Results of short-term from-scratch forecasting. All results are averaged across four forecasting horizon: $H \in \{12, 24, 48, 96\}$. The best and second-best results are highlighted in **bold** and underlined, respectively.

| Models | | ConFlux Ours | | DUET 2025 | | CycleNet 2024 | | TimeXer 2024 | | SOFTS 2024 | | iTransformer 2023 | | PatchTST 2023 | | Crossformer 2023 | | TimesNet 2022 | | DLinear 2022 | | SCINet 2022 | |
|---|---|---|---|---|---|---|---|---|---|---|---|---|---|---|---|---|---|---|---|---|---|---|---|
| Metrics | | MSE | MAE | MSE | MAE | MSE | MAE | MSE | MAE | MSE | MAE | MSE | MAE | MSE | MAE | MSE | MAE | MSE | MAE | MSE | MAE | MSE | MAE |
| PEMS03 | 12 | **0.061** | **0.162** | 0.064 | 0.166 | 0.070 | 0.173 | 0.066 | 0.172 | 0.064 | 0.165 | 0.071 | 0.174 | 0.099 | 0.216 | 0.090 | 0.203 | 0.085 | 0.192 | 0.122 | 0.243 | 0.066 | 0.172 |
| | 24 | **0.078** | **0.183** | 0.081 | 0.186 | 0.092 | 0.194 | 0.089 | 0.201 | 0.083 | 0.188 | 0.093 | 0.201 | 0.142 | 0.259 | 0.121 | 0.240 | 0.118 | 0.223 | 0.201 | 0.317 | 0.085 | 0.198 |
| | 48 | **0.110** | **0.217** | 0.114 | 0.222 | 0.129 | 0.229 | 0.136 | 0.247 | 0.114 | 0.223 | 0.125 | 0.236 | 0.211 | 0.319 | 0.202 | 0.317 | 0.155 | 0.260 | 0.333 | 0.425 | 0.127 | 0.238 |
| | 96 | **0.149** | **0.259** | 0.153 | 0.273 | 0.157 | 0.261 | 0.182 | 0.282 | 0.156 | 0.264 | 0.164 | 0.275 | 0.269 | 0.370 | 0.262 | 0.367 | 0.228 | 0.317 | 0.457 | 0.515 | 0.178 | 0.287 |
| | *Avg.* | **0.100** | **0.205** | 0.103 | 0.212 | 0.118 | 0.226 | 0.112 | 0.214 | 0.104 | 0.210 | 0.113 | 0.222 | 0.180 | 0.291 | 0.169 | 0.282 | 0.147 | 0.248 | 0.278 | 0.375 | 0.114 | 0.224 |
| PEMS04 | 12 | **0.066** | **0.164** | 0.079 | 0.181 | 0.078 | 0.186 | 0.074 | 0.178 | 0.074 | 0.176 | 0.078 | 0.183 | 0.105 | 0.224 | 0.098 | 0.218 | 0.087 | 0.195 | 0.148 | 0.272 | 0.073 | 0.177 |
| | 24 | **0.077** | **0.178** | 0.096 | 0.203 | 0.099 | 0.212 | 0.087 | 0.195 | 0.088 | 0.194 | 0.095 | 0.205 | 0.153 | 0.275 | 0.131 | 0.256 | 0.103 | 0.215 | 0.224 | 0.340 | 0.084 | 0.193 |
| | 48 | **0.095** | **0.202** | 0.114 | 0.226 | 0.133 | 0.248 | 0.110 | 0.214 | 0.110 | 0.219 | 0.120 | 0.233 | 0.229 | 0.339 | 0.205 | 0.326 | 0.136 | 0.250 | 0.355 | 0.437 | 0.099 | 0.211 |
| | 96 | 0.120 | **0.227** | 0.136 | 0.257 | 0.167 | 0.281 | 0.148 | 0.251 | 0.135 | 0.244 | 0.150 | 0.262 | 0.291 | 0.389 | 0.402 | 0.457 | 0.190 | 0.303 | 0.452 | 0.504 | **0.114** | **0.227** |
| | *Avg.* | **0.079** | **0.181** | 0.106 | 0.217 | 0.119 | 0.232 | 0.105 | 0.209 | 0.102 | 0.208 | 0.111 | 0.221 | 0.195 | 0.307 | 0.209 | 0.314 | 0.129 | 0.241 | 0.295 | 0.388 | 0.093 | 0.202 |
| PEMS07 | 12 | **0.054** | **0.149** | 0.060 | 0.156 | 0.062 | 0.162 | 0.057 | 0.152 | 0.057 | 0.152 | 0.067 | 0.165 | 0.095 | 0.207 | 0.094 | 0.200 | 0.082 | 0.181 | 0.115 | 0.242 | 0.068 | 0.171 |
| | 24 | **0.068** | **0.164** | 0.073 | 0.172 | 0.086 | 0.192 | 0.079 | 0.179 | 0.073 | 0.173 | 0.088 | 0.190 | 0.150 | 0.262 | 0.139 | 0.247 | 0.101 | 0.204 | 0.210 | 0.329 | 0.119 | 0.225 |
| | 48 | **0.088** | **0.188** | 0.096 | 0.201 | 0.128 | 0.234 | 0.099 | 0.191 | 0.096 | 0.195 | 0.110 | 0.215 | 0.253 | 0.340 | 0.311 | 0.369 | 0.134 | 0.238 | 0.398 | 0.458 | 0.149 | 0.237 |
| | 96 | 0.118 | 0.217 | 0.131 | 0.255 | 0.176 | 0.268 | **0.107** | **0.205** | 0.120 | 0.218 | 0.139 | 0.245 | 0.346 | 0.404 | 0.396 | 0.442 | 0.181 | 0.279 | 0.594 | 0.553 | 0.141 | 0.234 |
| | *Avg.* | **0.070** | **0.167** | 0.090 | 0.196 | 0.113 | 0.214 | 0.085 | 0.182 | 0.087 | 0.184 | 0.101 | 0.204 | 0.211 | 0.303 | 0.235 | 0.315 | 0.124 | 0.225 | 0.329 | 0.396 | 0.119 | 0.217 |
| PEMS08 | 12 | **0.062** | **0.158** | 0.072 | 0.168 | 0.082 | 0.185 | 0.075 | 0.176 | 0.074 | 0.171 | 0.079 | 0.182 | 0.168 | 0.232 | 0.165 | 0.214 | 0.112 | 0.212 | 0.154 | 0.276 | 0.087 | 0.184 |
| | 24 | **0.077** | **0.176** | 0.093 | 0.191 | 0.117 | 0.226 | 0.102 | 0.201 | 0.104 | 0.201 | 0.115 | 0.219 | 0.224 | 0.281 | 0.215 | 0.260 | 0.141 | 0.238 | 0.248 | 0.353 | 0.122 | 0.221 |
| | 48 | **0.101** | **0.206** | 0.123 | 0.217 | 0.169 | 0.268 | 0.158 | 0.248 | 0.164 | 0.253 | 0.186 | 0.235 | 0.321 | 0.354 | 0.315 | 0.355 | 0.198 | 0.283 | 0.440 | 0.470 | 0.189 | 0.270 |
| | 96 | **0.141** | **0.248** | 0.166 | 0.291 | 0.233 | 0.306 | 0.366 | 0.377 | 0.211 | 0.253 | 0.221 | 0.267 | 0.408 | 0.417 | 0.377 | 0.397 | 0.320 | 0.351 | 0.674 | 0.565 | 0.236 | 0.300 |
| | *Avg.* | **0.095** | **0.197** | 0.114 | 0.217 | 0.150 | 0.246 | 0.175 | 0.250 | 0.138 | 0.219 | 0.150 | 0.226 | 0.280 | 0.321 | 0.268 | 0.307 | 0.193 | 0.271 | 0.379 | 0.416 | 0.159 | 0.244 |

## Q. Limitation and Future Work

While ConFlux achieves strong performance in multivariate time series forecasting across zero-shot, fine-tuning, and from-scratch settings, it is currently evaluated primarily on forecasting tasks. Exploring its applicability to other time series analysis tasks may further broaden its practical impact. In addition, similar to existing time series foundation models, the scale of training data and model parameters remains constrained by the availability of large, high-quality multivariate time series datasets. Future work will focus on extending the ConFlux framework to a wider range of time series tasks and on leveraging larger and more diverse multivariate datasets to further improve generalization and scalability. These directions may help better understand the limits and potential of variate-centric foundation models for time series analysis.

# R. Additional Visualizations of Variate Sorting

We provide additional visualizations of the aggregated association matrix $A$ before and after variate sorting on multiple datasets. Before sorting, strong cross-variable dependencies are distributed irregularly across the matrix due to the arbitrary original variate order. After sorting based on the Fiedler vector, high association values become concentrated near the diagonal, forming a clear banded structure. This indicates that strongly related variates are placed adjacently, which improves locality in the variate dimension and better aligns with variate patching. These visual results further support that variate sorting reduces information interference when aggregating variates into patches and contributes to improved multivariate forecasting performance.

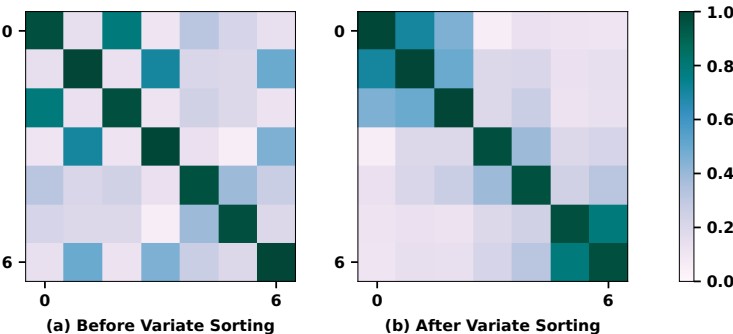

(a) Before Variate Sorting       (b) After Variate Sorting

*Figure 9.* ETTh1

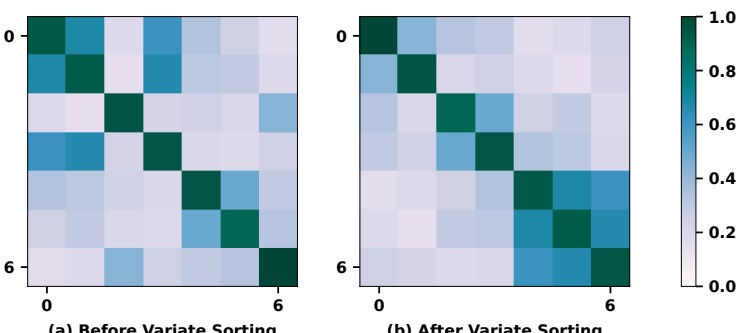

(a) Before Variate Sorting       (b) After Variate Sorting

*Figure 10.* ETTh2

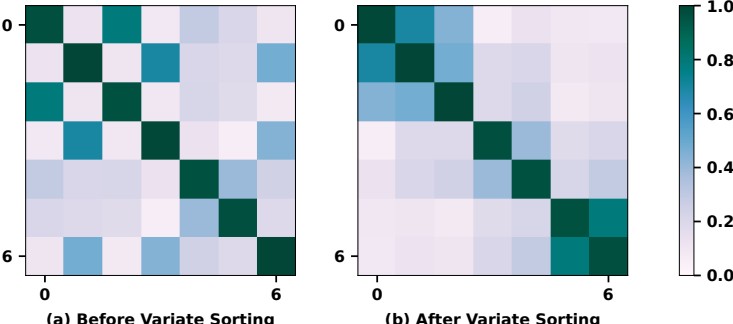

(a) Before Variate Sorting       (b) After Variate Sorting

*Figure 11.* ETTm1

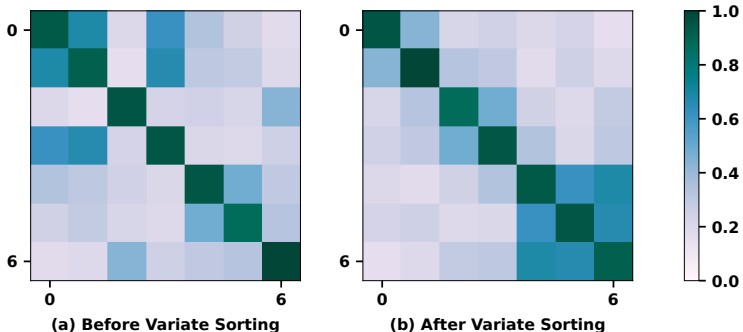

Figure 12. ETTm2

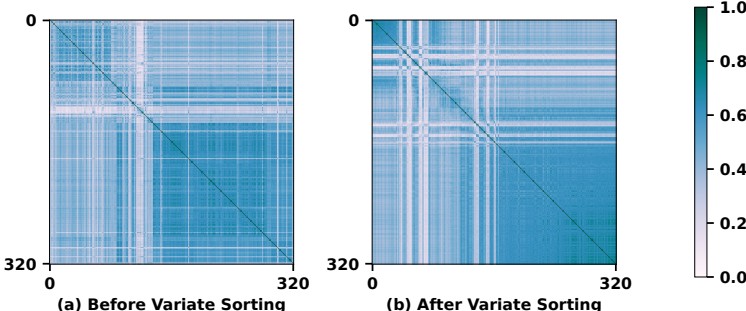

Figure 13. Electricity

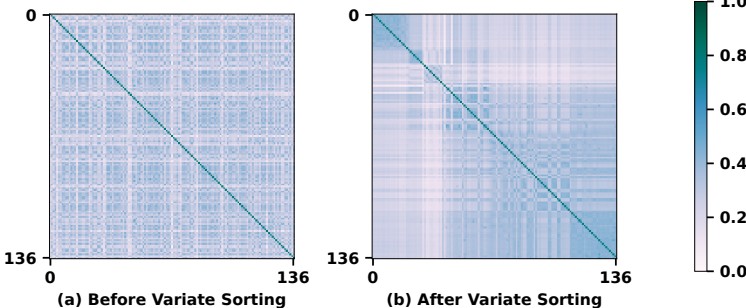

Figure 14. Solar

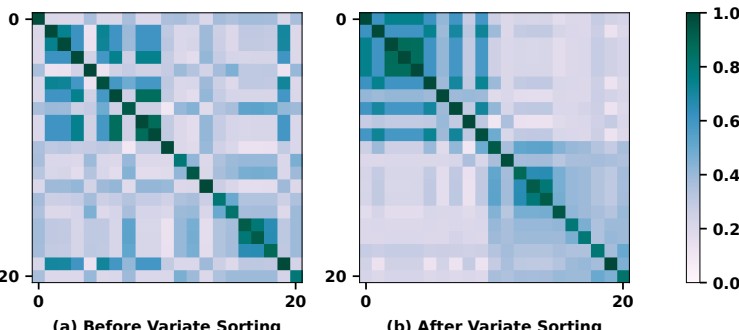

Figure 15. Weather

# S. Show Case

We present representative forecasting cases to qualitatively compare ConFlux with baseline methods. The showcases visualize predicted trajectories against ground-truth values on selected variates and horizons. Compared with baselines, ConFlux produces forecasts that better follow the overall trend and periodic patterns, while exhibiting reduced phase shift and amplitude distortion. These examples illustrate that the proposed variate sorting and variate patching enable more coherent cross-variable modeling, leading to more stable and accurate forecasts in practice.

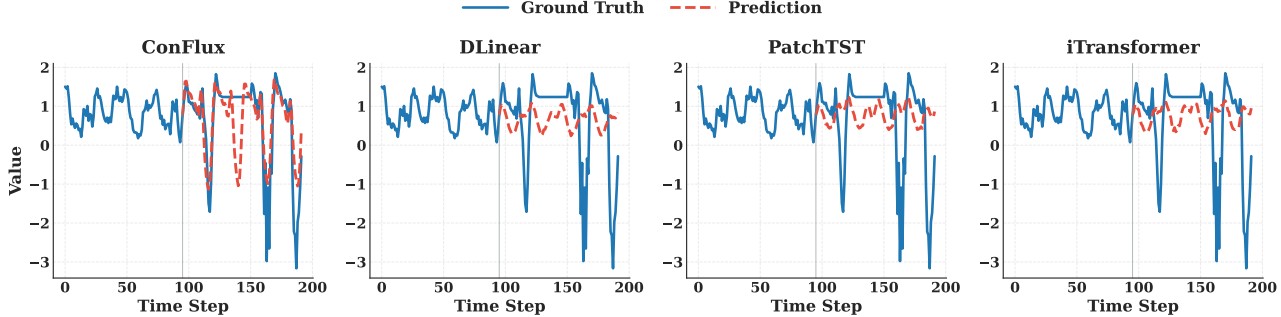

*Figure 16.* ETTh1

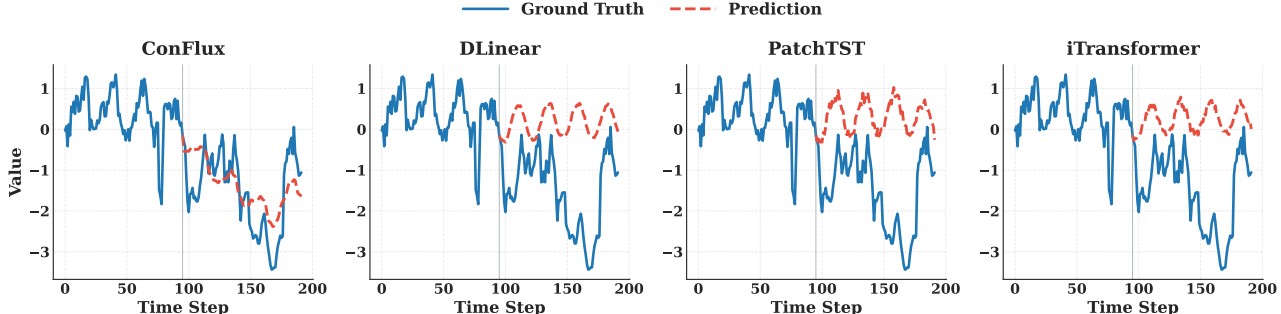

*Figure 17.* ETTh2

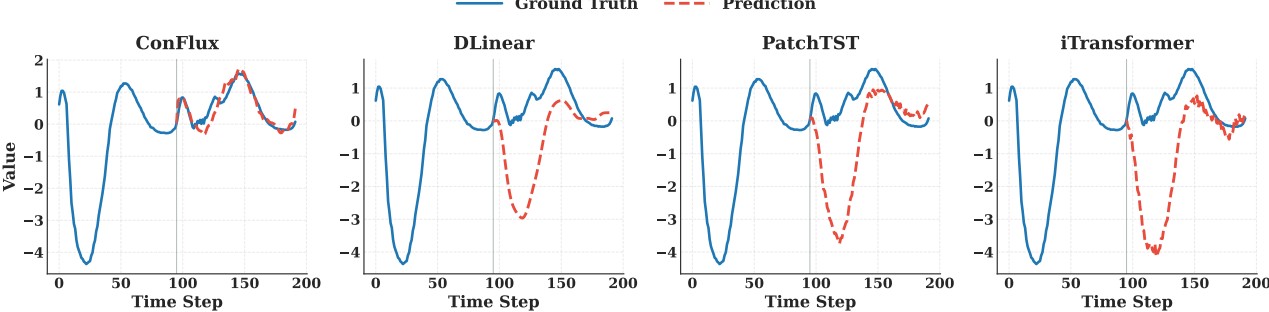

*Figure 18.* ETTm1

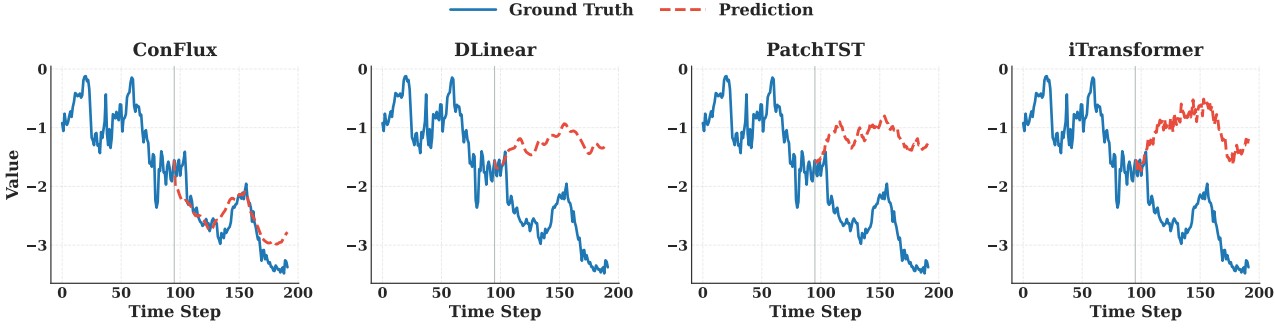

*Figure 19.* ETTm2

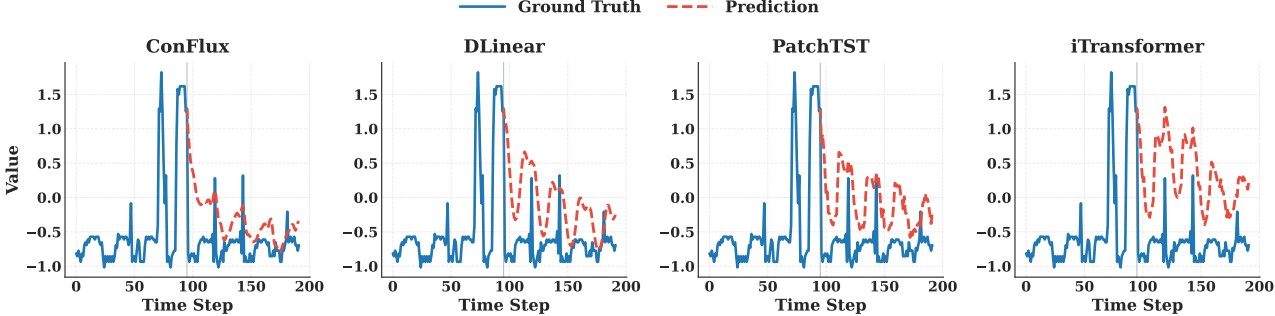

*Figure 20.* Electricity

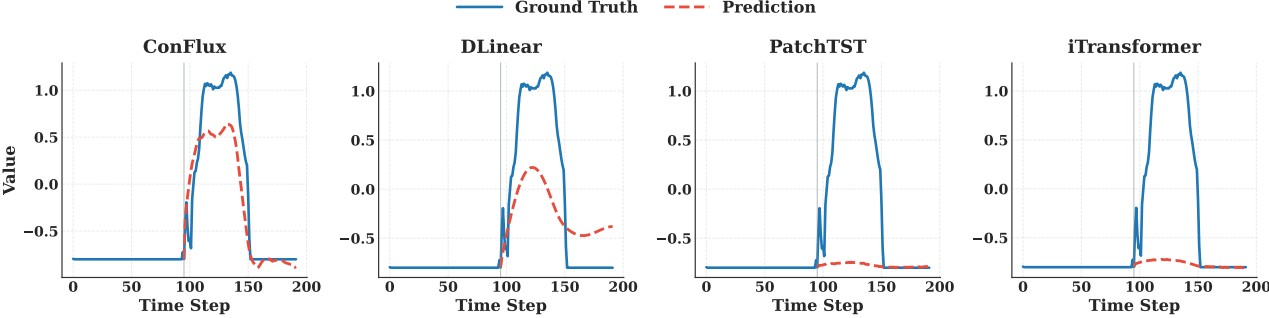

*Figure 21.* Solar

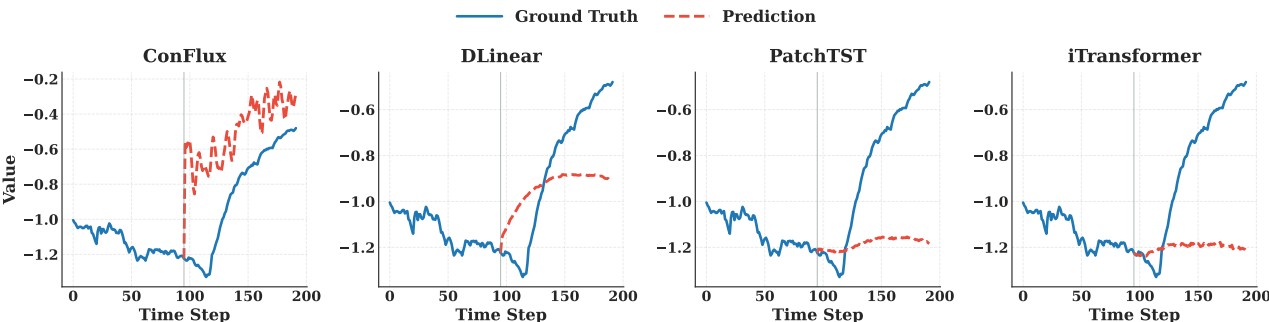

*Figure 22.* Weather

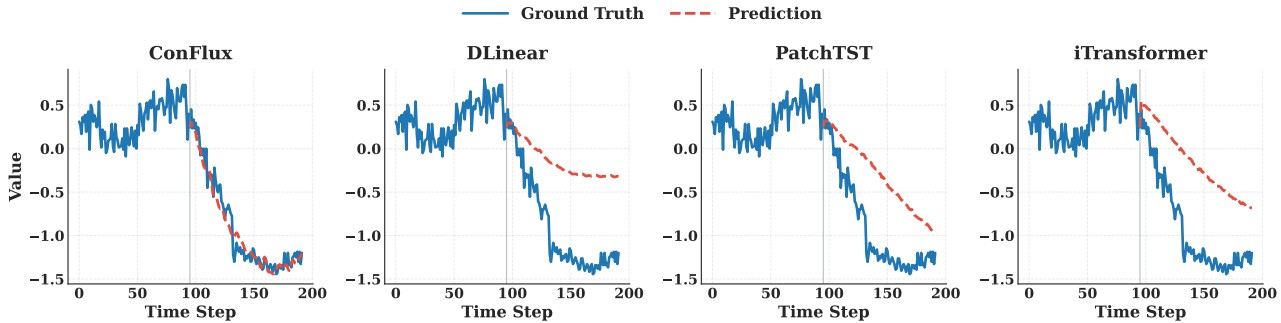

*Figure 23.* PEMS03

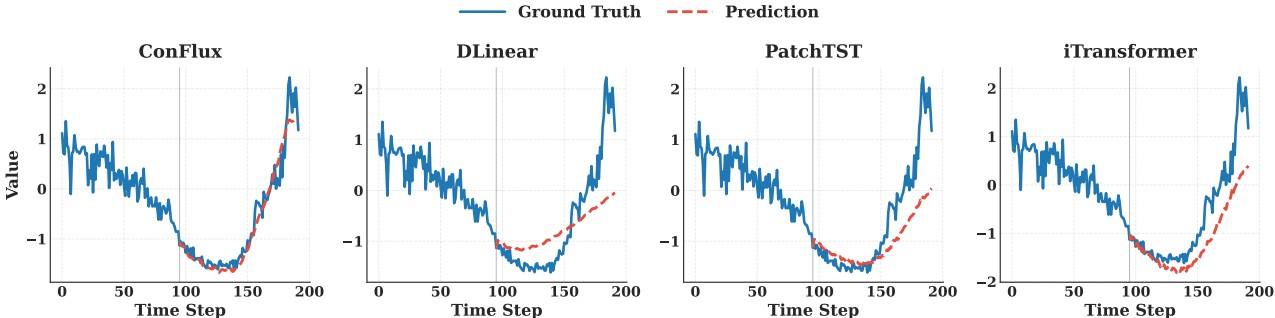

*Figure 24.* PEMS04

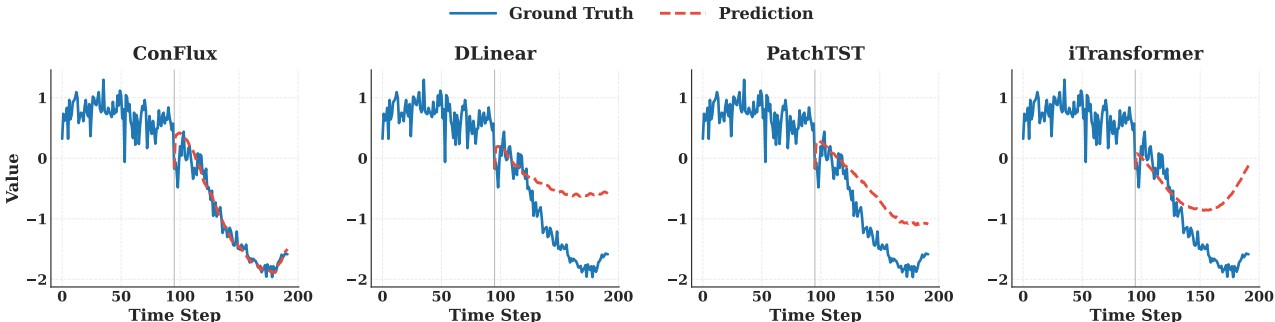

*Figure 25.* PEMS07

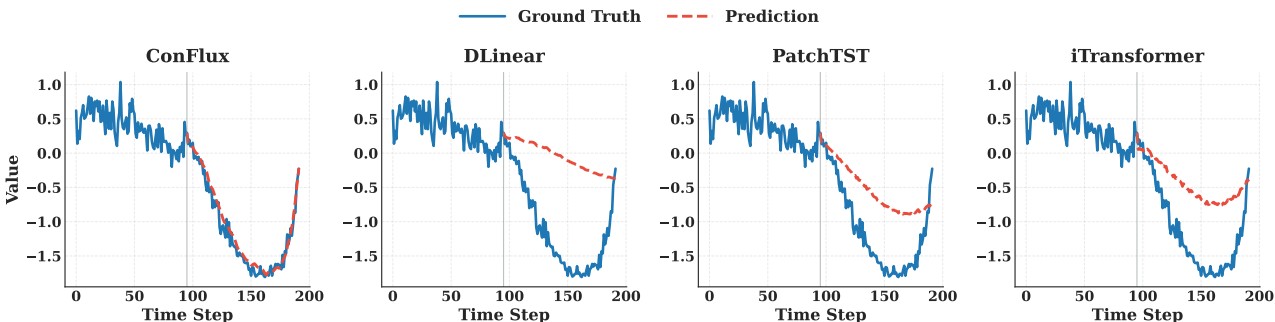

*Figure 26.* PEMS08

