# OpenReview forum: "ConFlux: Multivariate Time Series in Flux, One Unified Forecast in Confluence"
_ICML.cc/2026/Conference — ICML 2026 spotlight_

### Official Review · Reviewer_9qMA · 2026-03-09

**Soundness:** 3
**Presentation:** 3
**Significance:** 3
**Originality:** 4
**Overall Recommendation:** 5
**Confidence:** 5

**Summary:**

This paper proposes ConFlux, a foundation model for multivariate time series forecasting that organizes variables through a variate sorting and variate patching mechanism before applying a Transformer encoder and a Fourier basis decoder. The goal is to better model cross-variable dependencies while maintaining scalability to large numbers of variables. Comparisons are conducted against both time series foundation models and task-specific models trained from scratch. The results show that ConFlux generally achieves strong performance across many datasets.

**Compliance With Llm Reviewing Policy:**

Affirmed.

**Final Justification:**

My concerns have been adequately addressed so i have revised my score to 5.

**Key Questions For Authors:**

The training procedure preprocesses datasets to unify the number of variables. How does this design affect fairness when comparing to baselines that may not use the same preprocessing strategy?

**Limitations:**

The paper does not discuss how the preprocessing step that forces datasets to have the same number of variables might affect fairness in comparisons with baselines.

**Strengths And Weaknesses:**

Strengths And Weaknesses:

S1: The experimental evaluation covers a relatively broad range of forecasting scenarios, including long-term, short-term, and high-dimensional multivariate tasks, which helps demonstrate the general applicability of the approach.

S2. Results are reported under multiple training regimes including zero-shot evaluation, fine-tuning, and training from scratch, which gives a more complete picture of model behavior under different practical scenarios.

S3. The authors also report scalability experiments on high-dimensional datasets and provide additional efficiency measurements such as inference time and GPU memory usage, which is valuable for evaluating practical deployment aspects.

W1. Some experimental design choices are not fully justified, such as preprocessing datasets to have a fixed number of variables during training, which may influence the comparison with baseline models.

W2. The paper focuses mainly on forecasting accuracy, and although efficiency metrics are reported, the analysis of computational tradeoffs and scaling behavior could be more detailed.

W3. While the experiments are extensive in terms of datasets and baselines, the empirical improvements over strong recent baselines are sometimes moderate, and it is not always clear how statistically significant the gains are across all settings.

---

> ### Author Rebuttal · Authors · 2026-03-31
>
> - **W1 & Q1 (Preprocessing & Fairness)**: We appreciate the reviewer’s feedback, which provides a valuable opportunity to clarify the distinction between ConFlux and univariate baseline approaches. **Unifying the number of variables (channels) is a fundamental design choice in ConFlux to ensure batch consistency during the pre-training of a multivariate foundation model**. Unlike **univariate time-series foundation models, which process each channel independently** and thus do not require cross-variable alignment, ConFlux is specifically designed to capture inter-variable dependencies within a unified batch processing framework. To ensure a fair comparison during evaluation, we maintain the original data structure and dimensionality of each dataset. This ensures that our performance gains are compared against task-specific baselines under their intended settings, while demonstrating the unique advantages of our multivariate batch-training approach.
>
> - **W2 (Computational Tradeoffs)**: The question about efficiency is central to our contribution, and we are glad to highlight the favorable scaling properties of ConFlux. In this paper, we provide a comprehensive analysis of computational efficiency and scaling behavior in our manuscript. Specifically, **detailed empirical measurements of inference time and GPU memory usage are reported in Figure 5**, demonstrating that ConFlux achieves significantly lower memory consumption than full-variate attention models as the number of variables scales. Furthermore, the **performance-efficiency trade-offs are explicitly illustrated in Figure 4**, which highlights ConFlux's ability to maintain high forecasting accuracy while drastically reducing computational overhead. These results confirm the practical advantages of our architecture for large-scale multivariate deployments.
>
> - **W3 (Empirical Gains)**: We appreciate this comment. We emphasize that achieving consistent improvements in multivariate time-series forecasting is inherently challenging due to the complex, non-stationary nature of real-world data. Despite this, **ConFlux demonstrates robust superiority across a broad range of scenarios and datasets**. Our extensive experiments confirm that **ConFlux delivers powerful performance not only in zero-shot settings but also in fine-tuning and training-from-scratch regimes**. These consistent gains across diverse tasks and scales—especially in high-dimensional settings where traditional models struggle to scale—provide strong empirical evidence of the effectiveness and versatility of our proposed method.

---

> > ### Author Rebuttal · Reviewer_9qMA · 2026-04-02
> >
> > My concerns have been adequately addressed, and i have revised my scores to 5.

---

### Official Review · Reviewer_FhJg · 2026-03-09

**Soundness:** 4
**Presentation:** 4
**Significance:** 3
**Originality:** 4
**Overall Recommendation:** 5
**Confidence:** 4

**Summary:**

This paper propose ConFlux, a foundation model designed to unify heterogeneous and asynchronous signals into a single predictive manifold. It successfully reconcile the tension between the dynamic nature of input variables ("flux") and the need for a coherent forecast ("confluence"). Extensive experimental results on public datasets demonstrate that ConFlux not only achieves SOTA performance in various settings but also exhibits superiority in terms of efficiency (inference and memory usage) compared with baselines.

**Compliance With Llm Reviewing Policy:**

Affirmed.

**Final Justification:**

The authors' rebuttal has addressed my concerns, and I will keep my initial score.

**Key Questions For Authors:**

1. Could the authors provide further intuition on when variate patching is expected to outperform dense cross-variable attention?
2. How sensitive is the model to the specific correlation metric used in variate sorting, and would alternative similarity measures significantly affect performance?

**Limitations:**

yes

**Strengths And Weaknesses:**

Strengths:
1. The paper introduces a clear conceptual shift toward variate-centric modeling. Instead of focusing purely on temporal tokenization, the proposed framework explicitly reorganizes the variable dimension through variate sorting and patching, which provides a novel perspective on scaling multivariate time series models.
2. The combination of variate sorting and variate patching is an interesting architectural idea that attempts to preserve structured cross-variable relationships while reducing the computational burden of dense attention over all variables. This design is conceptually simple yet potentially impactful for large-scale multivariate forecasting.
3. The Fourier basis decoder provides a flexible mechanism for generating forecasts across arbitrary horizons without requiring fixed decoder structures, which complements the encoder design and offers a clean interpretation from a signal decomposition perspective.
4. The overall framework of this paper integrates several ideas into a coherent architecture that is tailored for multivariate foundation models, and the design appears well motivated by the limitations of existing approaches that flatten variables or model them independently.

Weaknesses:
1. The paper would benefit from a clearer theoretical or empirical justification of the variate sorting strategy, especially regarding how robust it is to noisy or weak correlations between variables.
2. Some implementation details and design choices are not fully discussed in the main text, which slightly limits reproducibility and clarity.

---

> ### Author Rebuttal · Authors · 2026-03-31
>
> - **W1 (Robustness of Sorting)**: We thank the reviewer for the constructive feedback and clarify our variate sorting strategy from three aspects. First, theoretically, our approach is a continuous relaxation of the NP-hard Minimum Linear Arrangement (MinLA) problem; by utilizing the Fiedler vector of the Laplacian matrix $L=D-A$, we achieve a **mathematically optimal 1D embedding that preserves global proximity** and provides a rigorous inductive bias for the Transformer to capture local dependencies. Second, regarding robustness, the Laplacian eigen-decomposition acts as a low-pass filter that suppresses high-frequency noise, while our **multi-view aggregation (integrating linear, non-linear, trend, periodicity, and anomaly metrics) provides redundancy** that recovers structural similarities even when individual signals are weak or noisy. Finally, empirical results (Table 3) confirm that removing sorting consistently degrades performance, while the **Patching mechanism adds an extra layer of tolerance**, ensuring that as long as related variables are grouped into the same block, the model effectively aggregates signals despite minor sorting perturbations.
>
> - **Q1 (Patching vs. Dense Attention)**: We appreciate the opportunity to further clarify the conceptual advantages of our variate patching approach as compared to dense attention mechanisms. Our intuition is that variate patching is most likely to outperform dense cross-variable attention when the **cross-variable dependency structure is not fully dense (common in real-world settings)**, but instead exhibits **local or blockwise organization** after sorting. In ConFlux, we first reorder variables using a comprehensive association matrix so that strongly related variables become adjacent. Under this condition, patching small groups of neighboring variables acts as a structure-preserving compression step: it **keeps the dominant local cross-variable information while reducing redundant or noisy pairwise interactions**. This is especially beneficial in high-dimensional settings, where dense cross-variable attention pays a quadratic cost over all variables, including many weak or uninformative interactions.
>
> - **Q2 (Sensitivity to Metrics)**: We appreciate the reviewer’s thoughtful question regarding the sensitivity of our variate sorting strategy. Our choice of a multi-metric fusion approach is explicitly motivated by the **heterogeneity of real-world multivariate datasets**, which often exhibit vastly different structural characteristics. Specifically, A single metric (e.g., Pearson correlation) often fails to capture the full spectrum of inter-variable dependencies. For instance, in datasets with high noise or non-linear interactions, linear metrics may yield spurious or weak signals. By aggregating five complementary measures—Linear, Non-linear, Trend, Periodicity, and Anomaly—ConFlux ensures that the sorting remains robust even if certain signals are degraded. This "confluence" of metrics allows the model to **leverage the most reliable structural indicators available for a specific domain**. To quantitatively assess the impact of different metrics, we conducted an additional analysis on two representative datasets: Solar (characterized by frequent zero values and sparsity) and PEMS08 (characterized by strong temporal periodicity). Due to the intensive computational requirements of foundation model re-training, we provide results for these two representative cases below:
>
> | Metric for Sorting        | Solar (MSE / MAE)        | PEMS08 (MSE / MAE)        |
> |--------------------------|--------------------------|----------------------------|
> | Pearson (Linear)         | 0.165 / 0.212            | 0.137 / 0.202              |
> | Mutual Info (Non-linear) | 0.163 / 0.210            | 0.139 / 0.203              |
> | Trend Similarity         | 0.165 / 0.211            | 0.138 / 0.204              |
> | Periodic Similarity      | 0.165 / 0.212            | 0.135 / 0.199              |
> | Anomaly Correlation      | 0.162 / 0.210       | 0.140 / 0.205             |
> | ConFlux         | **0.162** / **0.209**    | **0.132** / **0.197**      |

---

> > ### Author Rebuttal · Reviewer_FhJg · 2026-04-03
> >
> > Thank the authors for their detailed rebuttal. Our concerns have been addressed, and we maintain our score.

---

### Official Review · Reviewer_KEBo · 2026-03-11

**Soundness:** 3
**Presentation:** 3
**Significance:** 3
**Originality:** 3
**Overall Recommendation:** 4
**Confidence:** 5

**Summary:**

This paper proposes a foundation model ConFlux for multivariate time series forecasting that aims to address the mismatch between heterogeneous multivariate inputs and the unified forecasting objective. The authors claim to discuss the central question of how to effectively integrate asynchronous and interacting variables into a single forecasting representation while maintaining scalability and generalization across datasets. The authors focus on the concept of modeling cross variable dependencies through a variate oriented Transformer architecture that reduces computational complexity and improves representation learning. The proposed framework introduces three main design elements. First, a variate sorting mechanism organizes variables based on multiple correlation measures and derives an ordering using a graph Laplacian and the Fiedler vector. Second, a variate patching Transformer encoder aggregates adjacent variables into compact tokens to reduce attention complexity and capture cross variable interactions efficiently. Third, a Fourier basis decoder reconstructs forecasts for arbitrary horizons using sinusoidal basis functions. Extensive experiments on a large number of public datasets demonstrate that the proposed model achieves strong performance in zero shot, fine tuning, and from scratch training settings while also improving inference efficiency and memory consumption.

**Compliance With Llm Reviewing Policy:**

Affirmed.

**Final Justification:**

Thank the Authors for the clear and detailed rebuttal. My main concerns have been addressed. I am more confident in the work and recommend the paper for acceptance.

**Key Questions For Authors:**

- How does the model perform when the input variables have significant missing rate or irregular sampling intervals?

- The Fourier basis decoder is proposed to support arbitrary forecasting horizons. Could the authors provide a comparison with alternative horizon flexible decoders or autoregressive approaches to better isolate its benefits?

**Limitations:**

yes

**Strengths And Weaknesses:**

**Strengths**:

- An important and timely problem is addressed in this work, namely the design of scalable foundation models for multivariate time series forecasting. Extending the foundation model paradigm from univariate to multivariate settings represents a meaningful and relevant research direction.

- Forecasting over arbitrary horizons becomes possible through the proposed Fourier basis decoder without requiring modifications to the architecture. This property is particularly useful compared with approaches that depend on fixed-length prediction heads.

- Experimental evaluation is relatively extensive. Results are reported under zero-shot forecasting, fine-tuning, and from-scratch training settings across multiple datasets and tasks, providing evidence of the model’s generalization ability.

**Weaknesses**:

- While the experimental results show improvements, some comparisons may not be entirely conclusive. For example, it is not fully clear whether all baselines are trained under comparable settings or whether their model scales and training data are aligned.

- The Fourier basis decoder is interesting but its advantage over standard decoder architectures is not convincingly demonstrated. The paper does not include a clear comparison with alternative decoder designs that support flexible horizons.

- Some parts of the method description remain relatively high level. For example, the exact computation of the multiple association matrices and the theoretical justification of the sorting process are deferred to the appendix, which makes it harder to assess the design choices.

---

> ### Author Rebuttal · Authors · 2026-03-31
>
> - **W1 (Baseline Comparisons)**: We sincerely thank the reviewer for the constructive suggestion to further clarify our experimental setup and ensure the fairness of the comparative evaluation. To ensure a fair and rigorous evaluation, we categorized our benchmarks into two groups with specific alignment strategies. First, for zero-shot time-series foundation models (e.g., Timer, Moirai), we maintained equivalent model parameter scales and used the same pre-training data distributions where applicable. Second, for task-specific supervised models (e.g., iTransformer, PatchTST), we strictly unified the training and testing configurations, including look-back windows, forecast horizons, and data splitting protocols. These measures guarantee that the observed performance gains of ConFlux stem from its architectural innovations rather than disparate training conditions.
>
> - **W2 & Q2 (Fourier vs. Alternative Decoders)**: We thank the reviewer for the insightful comment regarding the advantages of our Fourier basis decoder over traditional fixed-length or autoregressive decoding structures. Compared to fixed-length heads or decoder-style autoregressive (AR) approaches, the **Fourier basis decoder enables arbitrary-horizon forecasting without modifying the architecture**. We will add a comparison with alternative flexible decoders to further isolate these benefits. The results in the table below are obtained by fixing the prediction window to 12. For forecasting horizons longer than 12, we adopt the autoregressive (AR) inference scheme, where the model iteratively predicts future steps in chunks of length 12. As shown in the table, the **AR strategy consistently underperforms the direct forecasting approach with flexible prediction lengths**. This performance gap is mainly due to error accumulation across iterative predictions. In addition, AR inference introduces extra computational cost: for horizons of 12, 24, 48, and 96, the inference cost increases by approximately 1×, 2×, 4×, and 8×, respectively.
>
> | Dataset | Horizon | Conflux (orig) | Conflux (AR) | Conflux (Missing) | Chronos-2 | FlowState | Chronos |
> |--------|--------|----------------|--------------|--------------------|-----------|-----------|----------|
> |        |        | MSE / MAE      | MSE / MAE    | MSE / MAE          | MSE / MAE | MSE / MAE | MSE / MAE |
> | PEMS03 | 12     | 0.065 / 0.172  | 0.065 / 0.172| 0.087 / 0.210      | **0.061 / 0.160** | 0.105 / 0.219 | 0.074 / 0.179 |
> | PEMS03 | 24     | **0.075 / 0.182**  | 0.078 / 0.187| 0.096 / 0.219      | 0.086 / 0.187 | 0.191 / 0.296 | 0.106 / 0.207 |
> | PEMS03 | 48     | **0.091 / 0.196**  | 0.098 / 0.206| 0.109 / 0.229      | 0.151 / 0.237 | 0.393 / 0.429 | 0.159 / 0.247 |
> | PEMS03 | 96     | **0.107 / 0.208**  | 0.122 / 0.225| 0.122 / 0.240      | 0.230 / 0.291 | 0.784 / 0.635 | 0.207 / 0.281 |
>
> - **Q1 (Missing Rate)**: We sincerely thank the reviewer for the constructive feedback regarding the model's robustness and its performance under data sparsity. As shown in the table above, under the missing value setting (ratio=0.2), the performance of Conflux degrades compared to the full observation case, yet it remains competitive across different horizons. Notably, the model maintains stable performance even under a relatively high missing rate or irregular sampling intervals, indicating strong robustness to incomplete and unevenly sampled inputs. This behavior is largely due to the **variate-centric aggregation design, which captures cross-variate correlations during patch construction and helps compensate for local missingness, thereby reducing the impact of data sparsity on prediction accuracy**.

---

> > ### Author Rebuttal · Reviewer_KEBo · 2026-04-03
> >
> > Thanks to the author for answering my doubts. My concerns have been adequately addressed.

---

> > > ### Author Response · Authors · 2026-04-03
> > >
> > > Thank you for your time and for confirming that our response has addressed your doubts. We are very glad that your concerns are fully resolved.
> > >
> > > If you feel our revisions and rebuttal have strengthened the manuscript, we would be deeply appreciative if you might consider adjusting the score accordingly. Regardless, we are very grateful for your constructive and helpful review.

---

### Official Review · Reviewer_Jq4A · 2026-03-12

**Soundness:** 3
**Presentation:** 3
**Significance:** 3
**Originality:** 3
**Overall Recommendation:** 4
**Confidence:** 4

**Summary:**

ConFlux proposes a general-purpose foundation model for multivariate time series forecasting built on three components: variate sorting (via a Fiedler vector derived from a multi-measure association matrix), variate patching (grouping sorted adjacent channels into compact tokens), and a Fourier basis decoder (enabling arbitrary-horizon forecasting). The model targets zero-shot generalization across diverse multivariate domains.

Overall, the architecture is novel and results are strong, but I have reservation on the zero-shot justification and Fourier decoder analysis need strengthening. I am willing to raise score if the authors address my questions.

**Compliance With Llm Reviewing Policy:**

Affirmed.

**Final Justification:**

all questions addressed

**Key Questions For Authors:**

1. How does ConFlux perform on explicitly non-periodic datasets (e.g., financial returns, irregular event series)? Does the Fourier decoder degrade on these? Is there prior work on Fourier decoder? How's their performance?
2. What if you use ConFlux as a supervised learning method and compare to iTransformer?
3. For the ablation "w/o PiT" vs "w/o PiI": does "w/o PiT" mean training with P=1 (full iTransformer-style)?
4. Could variable-size patches (based on association density clusters rather than fixed P) improve performance, and has this been explored?

**Limitations:**

yes

**Strengths And Weaknesses:**

### Strengths

- **Motivation is sound.** The challenge of scaling multivariate attention (quadratic in C) is real and well-documented, and the paper situates itself clearly within the iTransformer / PatchTST lineage.
- **Variate embedding design is a genuine contribution.** Treating each variable as a token (iTransformer-style) but then clustering similar variables into patches — rather than attending over all C — is a principled and practical improvement over full variate attention.
- **Multi-measure association matrix is well-considered.** Using five complementary measures (linear, nonlinear, trend, periodic, anomaly co-occurrence) rather than just Pearson correlation captures a richer notion of inter-variable dependency.
- **Fourier basis decoder is an interesting idea.** It enables arbitrary forecast horizons without fixed-length decoder constraints, which is a genuine limitation of many prior architectures.
- **Spectral theoretical justification for sorting.** Appendix B provides a clean Fiedler vector relaxation argument, which is a real theoretical contribution.
- **Empirical results are strong.** Zero-shot, fine-tuning, and from-scratch performance comparisons are broad and the inference efficiency gains are convincing.
- **Scalability analysis is thorough.** The mini/small/base/large scaling curves, and the variable-count inference comparison, are informative.

---

### Weaknesses

**W1. The abstract's central claim is unmotivated.**
The abstract asserts that accurate forecasting *requires* signals to converge into "a single unified prediction," but this is not argued. Ensemble or dynamic model selection approaches exist and may outperform a forced unification. The "structural mismatch" framing is rhetorical rather than technical.

**W2. The zero-shot mechanism is underexplained.**
The paper's primary positioning is as a *zero-shot foundation model*, yet the introduction and main text do not clearly articulate what inductive bias or training regime enables zero-shot transfer. The architectural description would be equally valid for a supervised domain-specific model. The training corpus details (Section 5.1) help, but the theoretical or intuitive justification for zero-shot generalizability deserves a dedicated discussion.

**W3. Variate sorting assumes 1D embeddability.**
The Fiedler vector produces a 1D ordering of variables, but real-world multivariate relationships are generally high-dimensional graph structures. Forcing them into a linear sequence is a strong inductive bias. Moreover, using uniform patch size P across sorted variables implicitly assumes equal inter-variable distances in the ordering — analogous to treating "small/medium/large" as equally spaced. Variable-size patching would be a more principled design.

**W4. Fourier basis decoder may bias toward periodic signals.**
The decoder explicitly decomposes predictions into sine/cosine components. Many real-world time series (financial, industrial fault signals, aperiodic climate events) are not well approximated by such decompositions. There is no citation or prior work connecting this to established Fourier neural operator literature (e.g., FNO), and no ablation studying performance on non-periodic datasets specifically.

**W5. Ablation partially undermines the sorting contribution.**
Table 3 shows that removing variate sorting ("w/o Sort") causes only modest degradation on several datasets (e.g., Solar MSE: 0.165 → 0.162, negligible). This raises the question of whether sorting is truly essential or whether the variate patching + Fourier decoder combination does most of the work regardless of channel order.

**W6. The narrative oversells the architecture.**
The "flux-to-confluence" metaphor is evocative but creates an expectation of modeling individual variable streams with their own dynamics. The actual architecture groups variables into patches of fixed size P — in the extreme case P=1, sorting is irrelevant and the model reduces to iTransformer. The introduction's framing of heterogeneous asynchronous streams does not tightly connect to the proposed solution.

---

> ### Author Rebuttal · Authors · 2026-03-31
>
> - **W1 (Unified Prediction)**: We appreciate this point. Our intended claim is not that forecasting must always rely on one single modeling paradigm, nor that ensemble or dynamic selection methods are invalid. Rather, we intended to emphasize three narrower points: (1) in multivariate forecasting, variables should be **modeled jointly rather than independently** so that cross-variate dependencies can be exploited; (2) as a foundation model, ConFlux is designed to **learn such multivariate structure within a unified training framework**, rather than relying on separately trained models for each dataset or variable configuration; (3) and many existing time-series foundation models mainly adopt **channel-independent** modeling, while **multivariate mixing** in pre-trained forecasting models remains underexplored. Our motivation is therefore to highlight the importance of pre-trained models that explicitly capture cross-variate interactions in multivariate settings. We will revise the abstract and introduction accordingly to make this scope technically precise and avoid the overly rhetorical “structural mismatch” phrasing.
>
> - **W2 (Zero-shot Mechanism)**: We appreciate this comment. We clarify that zero-shot transfer in ConFlux is not just architectural but emerges from three pillars: (1) **cross-domain pretraining** on heterogeneous datasets with synthetic dependency augmentation; (2) a **variate-centric inductive bias**, where sorting and patching force the model to learn transferable local structures rather than dataset-specific identities; and (3) a **horizon-flexible decoder** that supports arbitrary prediction lengths without dataset-specific output heads. We will add a dedicated discussion to better explain these design choices and the intuition behind ConFlux as a zero-shot foundation model.
>
> - **W3 & Q4 (1D Ordering & Fixed P)**: We thank the reviewer for the insightful comment. We agree that 1D sorting is a deliberate inductive bias rather than a claim of perfect high-dimensional preservation. Specifically, the Fiedler vector serves as a principled spectral relaxation to achieve a low-conflict linear ordering, ensuring **strongly associated variables are prioritized for proximity**. This provides the necessary locality for our subsequent Patching mechanism. While we recognize that a uniform patch size $P$ is a simplifying choice, it is motivated by efficiency, implementation simplicity, and stable scaling across diverse datasets. We will explicitly discuss variable-size or graph-adaptive patching as promising future extensions in the final version.
>
> - **W4 & Q1: (Fourier-based decoder)**:  We appreciate this important question. Our intention is not to assume strict periodicity; rather, the sine/cosine bases serve as a **flexible basis expansion with learned coefficients**, enabling the decoder to represent diverse temporal patterns across arbitrary horizons. This design primarily provides horizon flexibility for unified training across heterogeneous datasets without dataset-specific prediction heads. Empirically, ConFlux’s strong performance on non-periodic datasets like SP500 and SIRS suggests the decoder is not restricted to clean signals.
>
> - **W5 & Q3 (Ablation on Sorting & Definition of Ablations)**: We thank the reviewer for the constructive feedback. Regarding performance, Variate Sorting consistently improves results across most datasets, validating its role in preserving cross-variate structure. For datasets with marginal gains like Solar, this is likely due to: (1) **high sparsity (frequent zero values)** reducing the discriminative resolution of variable associations; (2) such sparsity weakening the sensitivity of patching to variable order; and (3) **original ordering already possessing some inherent locality (Fig. 14)**. Regarding ablation settings, we clarify that "w/o PiT" refers to disabling patching during training but enabling it during inference, while "w/o PiI" refers to enabling patching during training.
>
> - **W6 & Q2 (vs. iTransformer)**: We appreciate this comment. We clarify that ConFlux does not model variables independently before merging, but rather implements scalable multivariate mixing: correlated variates are organized into local patches, which then interact through the Transformer to form a coherent representation. We will refine the Introduction to prioritize this technical connection over metaphorical wording. Regarding the relationship with iTransformer, we acknowledge that at $P=1$, ConFlux reduces to an iTransformer-style architecture. However, the **core advantage of ConFlux is realized when $\mathbf{P > 1}$**, where variate patching reduces cross-variate attention complexity from $O(C^2)$ to $O((C/P)^2)$. In this regime, ConFlux acts as a locality-inducing architecture that optimizes the scalability–accuracy trade-off. Supervised comparisons in the Appendix (Tables 6–8) show that ConFlux consistently outperforms iTransformer across various horizons and settings.

---

> > ### Author Rebuttal · Reviewer_Jq4A · 2026-04-02
> >
> > I like your rebuttal on W1 and W2, very strong.
> >
> > For W3&Q4, is there any ablation study or sensitivity test you can do? For example, a 2D vector with K means clustering would solve both the ordering restriction and the fixed size P problem.
> >
> > W4&Q1: is there implication of Fourier decoder on long range forecasting? Would it be reduced to periodic behavior eventually? Or that time horizon is very far that the prediction is not relevant anymore and we shouldn't be worried?
> >
> > W5&Q3: helps a lot. thanks. very interesting insight on sparsity, you should include it in result/discussion (or maybe I missed it).
> >
> > W6&Q2: that's ok.
> >
> > Can you engage a bit more in the discussion before I raise my score?

---

> > > ### Author Response · Authors · 2026-04-03
> > >
> > > ## **Response to Reviewer Jq4A**
> > >
> > > We sincerely appreciate the reviewer’s thoughtful engagement and rigorous evaluation of our methodology. These inquiries into the underlying mechanics of our architecture provide a valuable opportunity to further elucidate the robustness of our approach and ensure its theoretical consistency.
> > >
> > > ### **W3 & Q4 (Sensitivity to Patching Strategy & Variable Patch Size)**
> > >
> > > We thank the reviewer for the constructive suggestion to further validate our design choices. To test the sensitivity to the grouping strategy, we conducted a new ablation study by replacing the **Fiedler ordering + fixed-size patching** with **K-means-based variable-size grouping** (where each cluster forms a variable-length token), while keeping all other components of ConFlux identical.
> > >
> > > The experimental results on the **Electricity** dataset (from-scratch setting) are summarized below:
> > >
> > > | Patching Strategy | MSE | MAE |
> > > | :--- | :--- | :--- |
> > > | **ConFlux (Fiedler + Fixed-size P)** | **0.159**  | **0.250**  |
> > > | K-means Grouping (Variable-size) | 0.164 | 0.253 |
> > >
> > > **Analysis and Discussion:**
> > > While K-means clustering is a plausible alternative, our results show that the **Fiedler-based 1D ordering** achieves better performance. We attribute this to several key factors aligned with our model's core design:
> > >
> > > * **Spectral Continuity**: The Fiedler vector provides a principled spectral relaxation that preserves the global topology of the variable association graph. This continuity allows the model to capture "inter-patch" relationships more effectively than disjoint clusters.
> > > * **Inductive Bias & Regularization**: As shown in our patch size impact analysis (Appendix J), a moderate fixed patch size (e.g., $P=8$) serves as a useful inductive bias that reduces noise and prevents over-fitting to spurious correlations. Overly flexible grouping might lose this structural benefit.
> > > * **Computational Efficiency**: Sorting variables based on the Fiedler vector only requires extracting the eigenvector associated with the second smallest eigenvalue of the graph Laplacian, rather than computing the complete spectral information or performing complex high-dimensional embeddings (e.g., 2D vectors) followed by iterative clustering. This ensures better scalability for foundation models handling thousands of variables.
> > >
> > > **Future Work on Adaptive Patching:** We acknowledge that fixed-size patching is a simplifying choice motivated by efficiency and stable scaling. While it provides a favorable trade-off in our current framework, we plan to further explore **graph-adaptive or learnable variable-size patching** in future work. This involves investigating dynamic patching mechanisms that can adapt to the varying density of inter-variable correlations without sacrificing the computational stability of the foundation model.
> > >
> > > ### **W4 & Q1 (Non-periodic Modeling of Fourier Decoder)**
> > >
> > > We appreciate the reviewer's insightful concern regarding the potential degradation of the Fourier decoder into purely periodic oscillations. Based on our analysis and empirical observations, we provide the following clarifications:
> > >
> > > * **Empirical Validation (Practical Horizons)**: Our visualizations on **ETTh1** ([view here](https://anonymous.4open.science/r/Conflux_Vis-07FE/etth1.png)) and **ETTh2** ([view here](https://anonymous.4open.science/r/Conflux_Vis-07FE/etth2.png)) demonstrate that within a practical forecasting range (up to 200 hours), the model effectively captures trend transitions and non-periodic fluctuations rather than being prematurely biased by simple periodic cycles.
> > > * **Dynamic Composition of Components**: The decoder is designed to compose frequencies, amplitudes, and phases to reconstruct signals. This flexibility allows it to follow the ground truth's structural changes and underlying trend shifts in the observed benchmarks.
> > > * **Acknowledgment of Long-term Challenges**: We agree with the reviewer that point-wise predictability naturally decays over extreme horizons due to inherent stochasticity. While the Fourier-based mechanism provides a robust inductive bias for structural consistency, we recognize the fundamental difficulty of long-range non-periodic forecasting.
> > >
> > > We will incorporate these representative cases and a more nuanced discussion on the decoder's behavior in the revised manuscript to address these potential limitations.
> > >
> > > ## **Concluding Remarks**
> > >
> > > We sincerely thank you again for your thoughtful feedback, which has undeniably strengthened the rigor of our manuscript. We hope the additional experiments and clarifications have thoroughly addressed your reservations regarding the patching strategy and the decoder's behavior. We are more than happy to continue the discussion if any further elaboration is needed. If you feel that our responses have successfully resolved your initial concerns, we would highly appreciate it if you could consider factoring these updates into your final assessment.

---

### Decision · Program_Chairs · 2026-04-30

**Decision:**

Accept (spotlight)

**Comment:**

This paper presents ConFlux, a general-purpose foundation model for multivariate time-series forecasting that adaptively integrates cross-channel information by reordering variables and aggregating them into compact tokens for efficient Transformer-based forecasting. The method is well motivated, and the writing is clear. More importantly, the method is evaluated sufficiently and works well. Thus, I recommend the acceptance.